# Structural basis of transmembrane coupling of the HIV-1 envelope glycoprotein

Alessandro Piai [1,5], Qingshan Fu [1,5], Yongfei Cai[2,3,5], Fadi Ghantous [4], Tianshu Xiao [2,3], Md Munan Shaik[2,3], Hanqin Peng[2], Sophia Rits-Volloch[2], Wen Chen [1], Michael S. Seaman[4], Bing Chen[2,3 ✉] & James J. Chou [1 ✉]

The prefusion conformation of HIV-1 envelope protein (Env) is recognized by most broadly neutralizing antibodies (bnAbs). Studies showed that alterations of its membrane-related components, including the transmembrane domain (TMD) and cytoplasmic tail (CT), can reshape the antigenic structure of the Env ectodomain. Using nuclear magnetic resonance (NMR) spectroscopy, we determine the structure of an Env segment encompassing the TMD and a large portion of the CT in bicelles. The structure reveals that the CT folds into amphipathic helices that wrap around the C-terminal end of the TMD, thereby forming a support baseplate for the rest of Env. NMR dynamics measurements provide evidences of dynamic coupling across the TMD between the ectodomain and CT. Pseudovirus-based neutralization assays suggest that CT-TMD interaction preferentially affects antigenic structure near the apex of the Env trimer. These results explain why the CT can modulate the Env antigenic properties and may facilitate HIV-1 Env-based vaccine design.

[1] Department of Biological Chemistry and Molecular Pharmacology, Harvard Medical School, 250 Longwood Avenue, Boston, MA 02115, USA. [2] Division of Molecular Medicine, Boston Children's Hospital, Boston, MA 02115, USA. [3] Department of Pediatrics, Harvard Medical School, 3 Blackfan Street, Boston, MA 02115, USA. [4] Center for Virology and Vaccine Research, Beth Israel Deaconess Medical Center, 330 Brookline Avenue, Boston, MA 02215, USA. [5] These authors contributed equally: Alessandro Piai, Qingshan Fu, Yongfei Cai. ✉email: bchen@crystal.harvard.edu; james_chou@hms.harvard.edu

HIV-1 envelope glycoprotein [Env; trimeric (gp160)$_3$, cleaved to (gp120/gp41)$_3$] catalyzes fusion of viral and target cell membranes leading to viral entry[1,2]. Binding of gp120 to receptor (CD4) and co-receptor (e.g., CCR5 or CXCR4) triggers a cascade of refolding events in gp41 that promote membrane fusion[1,3,4]. The prefusion conformation of Env trimer is the state recognized by most broadly neutralizing antibodies (bnAbs)[5–7], and thus considered a major vaccine target. Several studies demonstrated that alterations of its membrane-related components, including the transmembrane domain (TMD) and cytoplasmic tail (CT), can reshape the antigenic structure of the Env ectodomain exposed outside of viral membrane[5,8,9], suggesting that there are intricate interconnections among them.

The intact HIV-1 Env has been visualized on the surface of virion at modest resolutions by cryo-electron tomography (cryo-ET)[10], revealing the trimeric organization of gp120 and a part of gp41, but leaving the TMD and CT regions completely unresolved. Its ectodomain density is in overall agreement with subsequent high-resolution structures, determined by X-ray crystallography and cryo-electron microscopy (cryo-EM), of a soluble form of the Env trimer stabilized by a disulfide crosslink between gp120 and gp41[11–13]. Other cryo-EM structures of detergent-solubilized Env constructs with or without the CT has been reported[14–16], but the MPER, TMD, and CT regions are all disordered in detergent micelles that probably failed to mimic a real membrane. We have recently determined the NMR structures of the trimeric MPER and TMD reconstituted in bicelles that mimic a lipid bilayer[8,9]. In particular, the MPER matches well the low-resolution cryo-ET density near the membrane of the unliganded viral Env, suggesting the MPER conformation observed by NMR is consistent with the structure of prefusion Env on the virion[9].

HIV-1 Env and related lentiviral fusion proteins have an unusually long CT (~150 residues), which has been implicated in Env cellular trafficking, as well as incorporation into virions[17,18]. The HIV-1 Env CT can be divided into distinct regions based on their biophysical properties (Supplementary Fig. 1a): a loop, commonly known as the Kennedy sequence (KS), followed by three segments predicted to form amphipathic α-helices, named lentivirus lytic peptide 2 (LLP2), LLP3, and LLP1[17]. Earlier studies suggested that the CT forms three membrane-bound amphipathic helices in an extended conformation[19,20]. These structures are very informative about the secondary structures of the CT, but they fall short of explaining how truncation in the CT can influence the antigenic structure of the ectodomain on the opposite side of the membrane[5,21,22]. In this study, we used NMR to obtain high-resolution information of the HIV-1 Env CT folding in the context of the TMD and lipid bilayer. We find that the CT adopts a structure different from the previous model[19] that can explain the physical coupling between the CT and the ectodomain.

## Results

**Identification of a suitable TMD–CT fragment for structural investigation.** To prepare an NMR sample suitable for structural analysis, we designed a protein construct derived from a clade D HIV-1 isolate 92UG024.2 (residues 677–788) that encompasses the TMD and a portion of the CT containing the KS and the LLP2. We first mutated the palmitoylation site C764 to serine (C764S) to avoid non-physiological disulfide formation. We then found that the KS (residues 710–738) was completely unstructured according to the NMR data and that removal of its central region (residues 726–736) did not affect the protein structure (Supplementary Fig. 2). This region was thus deleted to reduce NMR signal overlap. The final NMR construct, designated

TMD–CT$^{LLP2}$, included residues 677–725 and 737–788. TMD–CT$^{LLP2}$ is trimeric in DMPC–DHPC bicelles with $q = 0.5$ (Supplementary Fig. 1b). Comparison of the NMR spectrum of the bicelle-reconstituted TMD–CT$^{LLP2}$ with those of the MPER–TMD (residues 660–710)[9] and the TMD (residues 677–716)[8] used in previous studies showed that the peaks corresponding to the TM core (residues 685–700) are almost superimposable (Supplementary Figs. 1c and 3a), indicating that its structure is essentially identical in all three overlapping constructs. Therefore, the CT did not disrupt the TMD structure.

**NMR structure of the TMD–CT$^{LLP2}$ trimer in bicelles.** The structure of the trimeric TMD–CT$^{LLP2}$ in bicelles was determined using mainly two types of NMR-derived structural restraints. One is the inter-chain proton–proton distance, derived from nuclear Overhauser effects (NOEs) using previously established protocols[9,23] (see the Methods section and Supplementary Fig. 4). The other is the plane restraint (confinement of an assigned atom in a plane)[24], derived from NMR-based membrane partition analysis of the protein in bicelles[25]. The plane restraints are rarely used in NMR structure determination but useful here for constraining the CT segments that reside mostly on the plane of the bicelle. The residue-specific plane restraints were derived by (1) measuring solvent and lipophilic paramagnetic relaxation enhancement (PRE) amplitudes along the bicelle normal using the paramagnetic probe titration (PPT) method[25], and (2) calibrating the PRE amplitudes against the established structure and membrane partition of the TMD (residues 678–710)[8,26] (see Methods and Supplementary Fig. 5). The final structure of the TMD–CT$^{LLP2}$ trimer was independently validated by inter-chain PRE analyses of multiple site-directed spin-labeling (Supplementary Fig. 6).

The structure of the TMD–CT$^{LLP2}$ trimer in bicelles shows a novel membrane protein fold in which the CT winds around the TMD (Fig. 1a). The LLP2 is not a long continuous helix as previously proposed[19]; instead it forms two helices, designated H1 (residues 741–764) and H2 (residues 769–786), adopting a ring-like structure around the C-terminal region of the TMD trimer. The three H1s make direct contacts with the TMD trimer to form an inner ring around the TMD (Fig. 1a, b). The CT–TMD interactions involve the H1 nonpolar residues (L748 and L755), which form a hydrophobic cluster with TM residues (V701, L704, V705, and V708) of the same chain, and the polar H1 residue D759, which can make a salt bridge with the TM residue R709 of the neighboring chain. These residues thus collectively mediate the specific association between the CT ring and the TMD trimer. Indeed, hydrogen–deuterium (H–D) exchange data confirmed that residues 704–706, which showed fast H–D exchange in the TMD construct[26], are protected in the TMD–CT$^{LLP2}$ sample, as expected (Supplementary Fig. 7). The three H2s fold around the inner ring at its periphery (Fig. 1a, c). This folding appears to be driven mainly by hydrophobic interactions involving H2 residues (V778, I781, and V782) of one chain and H1 residues (I746 and V749) of the neighboring chain. Based on the membrane partition data (Fig. 1d), H1s and H2s all reside in the headgroup region of the lipid bilayer. The palmitoylation site C764 (S764 in this construct) faces the lipid bilayer interior, consistent with its role in anchoring the LLP2 to the membrane.

**A model of the MPER–TMD–CT$^{LLP2}$ in lipid bilayer.** Since the TM core (residues 685–700) has the same structure in the TMD, MPER–TMD, and TMD–CT$^{LLP2}$ samples (Supplementary Figs. 1c, 3a, and 8), it can guide the creation of a composite model containing all three components. As such, we merged the NMR

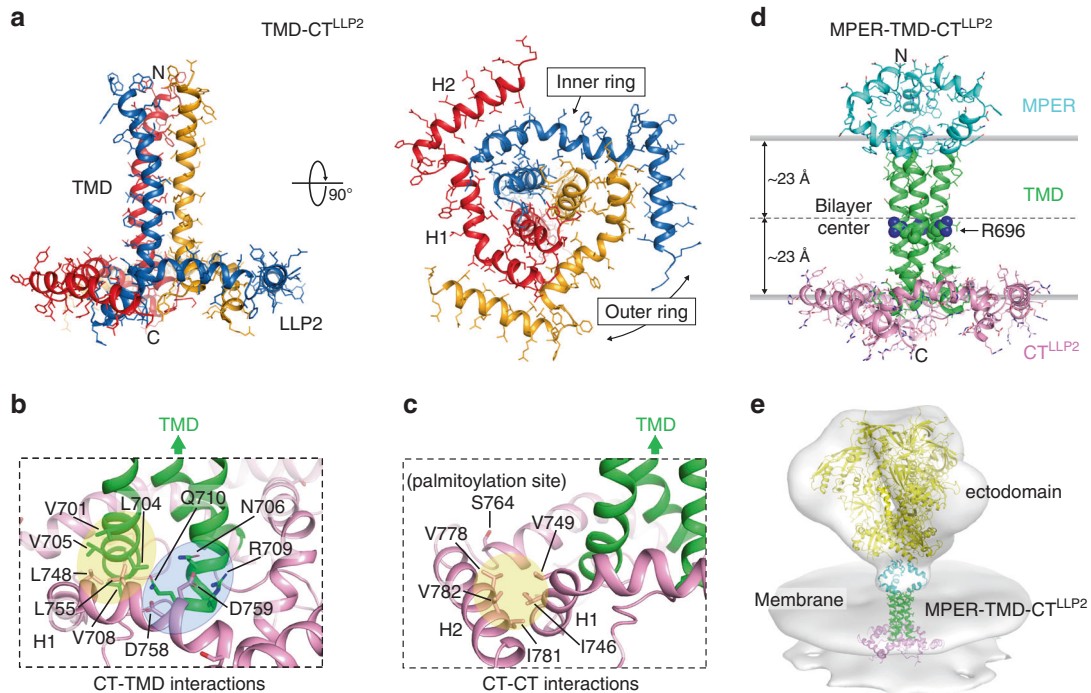

**Fig. 1 Structures of TMD–CT$^{LLP2}$ and MPER–TMD–CT$^{LLP2}$ trimers in bicelles. a** Ribbon representation of the TMD–CT$^{LLP2}$ average structure from the calculated ensemble. The unstructured KS (residues 711–736) is omitted for clarity. The H1 and H2 helices forming the inner and outer rings of the CT baseplate, respectively, are indicated in the bottom view (right). **b** Close-up view of the residues establishing the CT–TMD interactions. Hydrophobic and hydrophilic interactions are shaded in yellow and light blue, respectively. The TMD and the CT are shown as green and pink ribbons, respectively. **c** Same as (**b**) but for the CT–CT interactions. The palmitoylation site C764 (S764 in our construct) faces the lipid bilayer interior. **d** Ribbon representation of the merged MPER–TMD–CT$^{LLP2}$ model showing the MPER, TMD, and CT$^{LLP2}$ in cyan, green, and pink, respectively. The placement of the structure in the lipid bilayer was determined experimentally using the PPT method[25]. The conserved intramembrane R696 is represented as spheres. **e** Fit of the MPER–TMD–CT$^{LLP2}$ model and the structure of the SOSIP Env trimer (yellow; pdb ID: 5T3Z[61]) into the low-resolution EM density (gray) of the HIV-1 Env trimer on the virion surface by cryo-electron tomography[10] (Env trimer EMDB ID: EMD-5019; viral membrane EMDB ID: EMD-5020).

restraints from the three samples and constructed a model of the MPER–TMD–CT$^{LLP2}$ (residues 660–788) (Fig. 1d, e) in the context of lipid bilayer, as described in the Methods section. The model, which accounts for ~75% of the entire membrane region of the Env, shows that the membrane-related components all form well-structured trimeric assemblies capable of relaying structural perturbation from one component to another across the membrane.

**Physical interaction between the CT and the TMD.** To investigate the propensity of the CT to oligomerize and interact with the TMD, we split the TMD–CT$^{LLP2}$ construct into two fragments: one consisting only of the H1 and H2 helices (residues 737–788; designated CT$^{LLP2}$), and the other including the TMD and the KS (residues 677–725; designated TMD–KS) (Fig. 2a). When reconstituted in bicelles, the CT$^{LLP2}$ did not trimerize according to the OG-label analysis[27] (Supplementary Fig. 9a). In contrast, when reconstituted in the presence of the TMD–KS, the CT$^{LLP2}$ trimerized (Supplementary Fig. 9b) and folded around the TMD as shown by the PRE analysis (Fig. 2). In addition, to evaluate the structural impact of weakening the interaction between CT and TMD, we introduced five mutations in the CT H1 including L748S, L755S, D758A, D759A, and S762A. The mutant (designated CT2-tmd, also used in the antigenicity studies below) appeared to have maintained the overall trimeric structure of the wild type in bicelles as the NMR resonances of the TMD and H2 did not change significantly (Fig. 3a). The conformational stability of the mutant was examined by inter-chain PRE analysis, as was done in Supplementary Fig. 6b. Comparison of residue-specific PREs to that measured under identical condition for the

TMD–CT$^{LLP2}$ showed that the TMD PRE values of the mutant were reduced by as much as 30% (Fig. 3b). Since the two samples carried the same spin-label at C764, the results indicate that the H1 mutations above indeed had weaken the CT–TMD interaction and probably loosened the CT baseplate. The above independent experiments collectively suggest that the specific TMD–CT interactions drive spontaneous formation of the CT ring around the TMD.

**Interaction between the CT and TMD influences Env antigenicity.** In an earlier study, we showed that truncation or deletion of the CT diminishes Env binding to trimer-specific bnAbs that target the epitopes near the apex of the Env trimer[5,28–30]. With the new CT structural information, we now probe this effect with CT mutations designed to disrupt CT–TMD or CT–CT interactions shown in Fig. 1. The effect of the mutations on the antigenic structure of the Env ectodomain was evaluated by a pseudovirus-based neutralization assay[31], using bnAbs VRC01 (CD4 binding site)[32], PG9, PG16, and PGT145 (trimer-specific)[33,34], as well as non-neutralizing or strain-specific neutralizing antibodies, including b6 (CD4 binding site)[32], 3791 (V3)[35], and 17b (CD4-induced)[36] (mutant list and results in Supplementary Tables 1 and 2, respectively; epitope mapping is shown in Supplementary Fig. 10). Although virus neutralization is not a direct measure of antibody binding to the Env as the process could be influenced by other factors such as possible CT-matrix protein (MA) interaction and membrane fusion kinetics, earlier studies have shown that loss of neutralization is overall correlated with the loss of Env binding to bnAbs[5,37]. We generated 28 Env mutants using the sequence of a

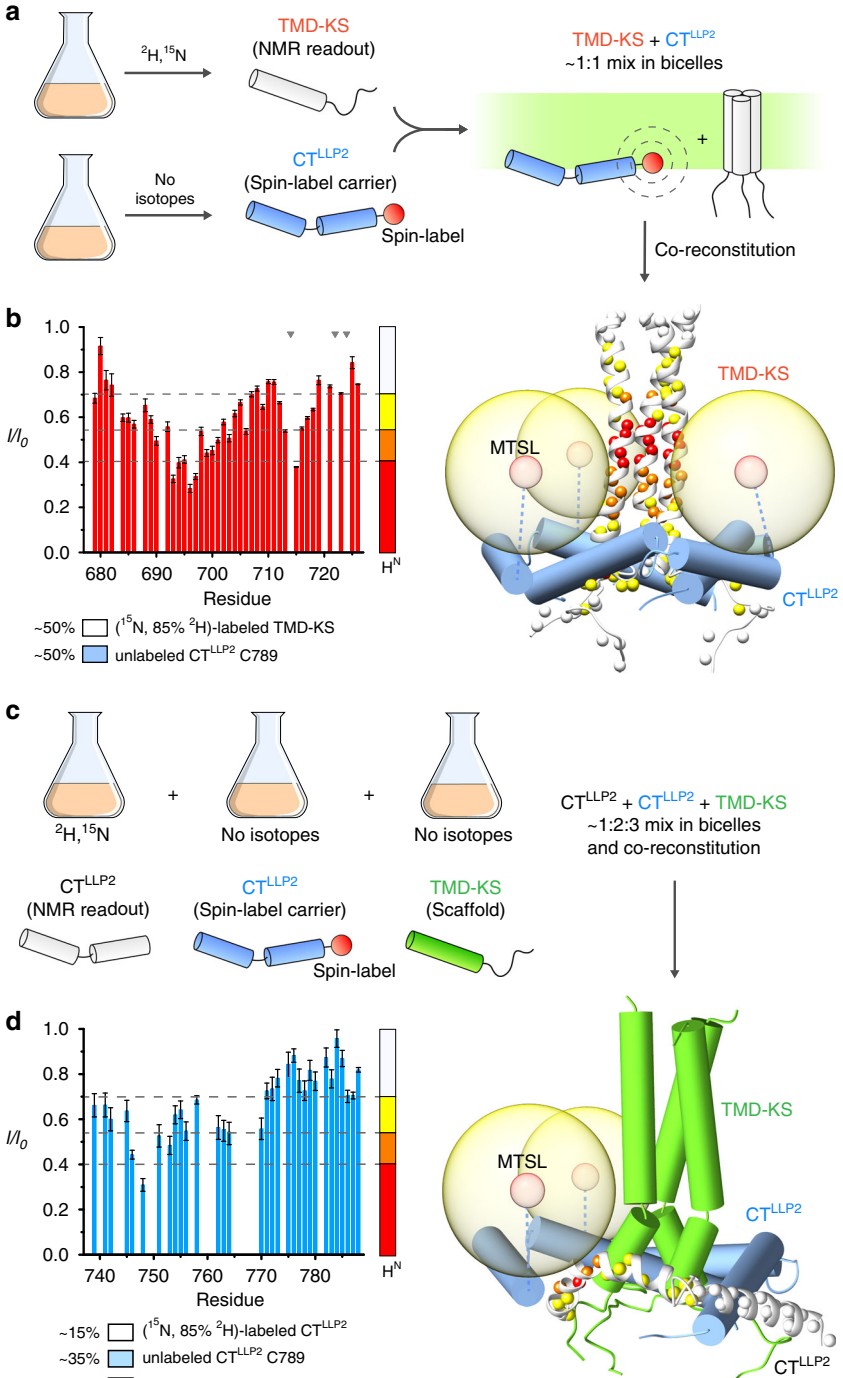

**Fig. 2 Independent evidence of CT$^{LLP2}$–TMD interaction. a** Schematic illustration of the sample preparation strategy used for intermolecular PRE analysis. TMD–KS and CT$^{LLP2}$ (C789) are expressed separately: TMD–KS is isotopically-enriched for NMR readout while CT$^{LLP2}$ carries the spin-label (at C789). After purification, the two segments are mixed at ~1:1 molar ratio and co-reconstituted in bicelles. Upon interaction, the TMD–KS is expected to experience PRE generated by the CT$^{LLP2}$ spin-label. **b** Residue-specific PRE ($I/I_0$) of ($^{15}$N, 85% $^2$H)-labeled TMD–KS mixed with MTSL-labeled CT$^{LLP2}$ (left). Error bars represent the uncertainty derived from cross-peaks signal to noise. Missing bars are due to prolines (indicated by gray triangles) or overlapping residues. The horizontal dash lines mark the four PRE regimes used to map the PREs onto the protein structure (right). The TMD–KS and the CT$^{LLP2}$ are shown as white ribbons and blue cylinders, respectively. **c** Labeling scheme for probing intermolecular CT$^{LLP2}$–CT$^{LLP2}$ interaction. CT$^{LLP2}$ (white) is isotopically-enriched for NMR readout while CT$^{LLP2}$ (blue) carries the spin-label at C789, and TMD–KS (green) serves as scaffold. After purification, the three proteins are mixed at ~1:2:3 molar ratio, respectively, and co-reconstituted in bicelles. **d** Residue-specific PRE ($I/I_0$) of ($^{15}$N, 85% $^2$H)-labeled CT$^{LLP2}$ mixed with MTSL-labeled CT$^{LLP2}$ and scaffold TMD–KS (left). Error bars represent the uncertainty derived from cross-peaks signal to noise. Missing bars are due to overlapping residues. The horizontal dash lines mark the four PRE regimes used to map the PREs onto the protein structure (right). Source data are provided as a Source data file.

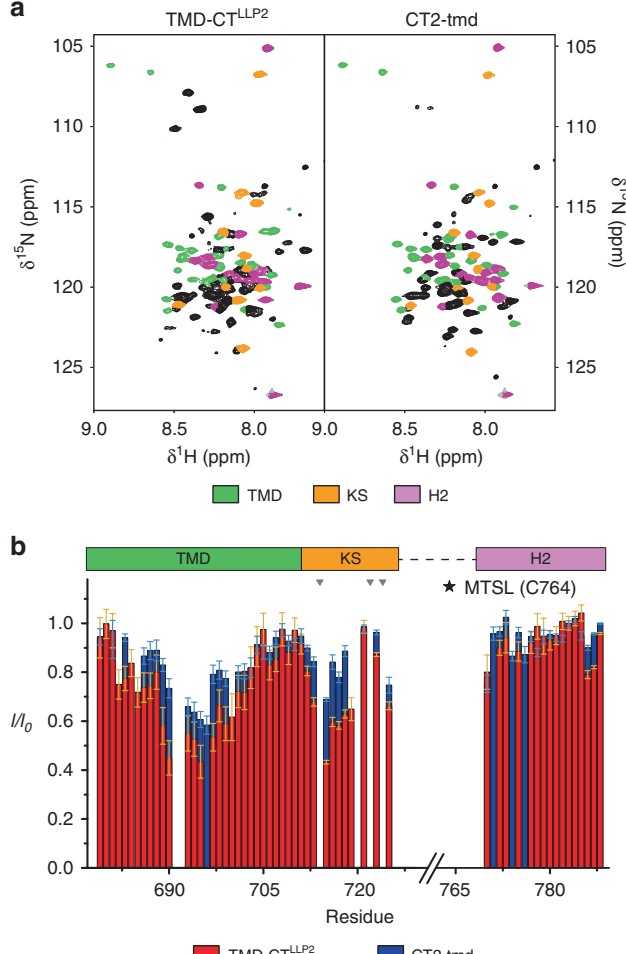

**Fig. 3 Destabilization of the TMD–CT interactions loosens the CT baseplate. a** Comparison of the 2D $^1$H-$^{15}$N TROSY-HSQC spectra of the TMD–CT$^{LLP2}$ (left) and CT2-tmd (right), showing strong agreement among the two constructs of the chemical shift of residues 679–710 (TMD), 711–725 (KS), and 770–788 (H2) highlighted in green, orange, and pink, respectively. Bearing the mutation site, H1 exhibited strong chemical shift changes (black cross-peaks) and thus was not further analyzed. **b** Comparison of residue-specific PRE ($I/I_0$) of TMD–CT$^{LLP2}$ (red) versus those of CT2-tmd (blue). Each sample was prepared mixing ~1:1 ($^{15}$N, 85% $^2$H)-labeled protein (NMR readout) with unlabeled (and thus "NMR-invisible") protein carrying the MTSL spin-label at position S764C, following the strategy summarized in Fig. 2a (see the Methods section for details). Error bars represent the uncertainty derived from cross-peaks signal to noise. Missing bars are due to prolines (indicated by gray triangles) or overlapping residues. The position of the paramagnetic tag is marked by a star. Source data are provided as a Source data file.

clade A isolate 92UG037.8, and we further characterized 11 of them (indicated in Supplementary Table 1) which showed significant phenotype in the antibody neutralization assay. When expressed in 293T cells, these mutants produced comparable levels of Env, with similar extents of cleavage between gp120 and gp41 (Supplementary Fig. 11). In particular, none of these mutations in the CT region have led to dramatic increase in cell-surface levels (Supplementary Fig. 12). At a low Env expression level to mimic the low surface density on HIV-1 virions[38], only the mutant TMD-ct showed a moderate reduction in cell–cell fusion activity, but still at ~50% of that of the wild-type Env (Supplementary Fig. 13a). At a high Env expression level, the fusion activity of all the mutants was essentially indistinguishable

from that of the wild type (Supplementary Fig. 13b), suggesting the high cell-surface density of Env can compensate for defects in membrane fusion caused by the CT mutations, as observed for mutations in the TMD and MPER previously[8,9]. When these mutations were introduced into pseudoviruses, there were moderate changes in Env incorporation for mutants V749K and A756N, and in processing for those with large-scale mutations, including the mutant TMD-ct (Supplementary Fig. 14), consistent with the role of the CT in Env incorporation[39]. In addition, the mutant F774N showed substantially increased viral infectivity (165%) while the rest of mutants have impaired infectivity, ranging from 10 to 63% of that of the wild-type Env, suggesting that the CT can both positively and negatively modulate the efficiency of viral entry (Supplementary Fig. 15).

As expected, the wild-type Env is neutralized by VRC01, PG9, PG16, and PGT145, but resistant to b6, 3791, and 17b (Fig. 4a). The single mutants all showed neutralization patterns very similar to that of wild-type Env (Supplementary Fig. 16). Among the four double mutants, the two containing D759R showed minor but noticeable reduction (~20%) in sensitivity to the trimer-specific bnAbs. The triple mutant (L702S-A756N-D759R) showed very significant resistance (~40%) to the trimer-specific bnAbs, especially for PG9 and PGT145 (Fig. 4b). These results suggest the important role of D759 in interacting with either R709 or N706 of the TMD (Fig. 1b), while underlining the nature of the CT structure being resistant to small changes. Larger scale mutagenesis of the CT–TMD interface resulted in mutants consistently less sensitive to the trimer-specific bnAbs, but remained totally resistant to the non-neutralizing antibodies (Fig. 4c and Supplementary Fig. 16). As a negative control, the mutant (A756N-L760S-L763N-C764E), containing mutations on the other side of the H1 helix, did not show any phenotype (Fig. 4d). Finally, hydrophobicity/hydrophilicity swapping of the key TMD residues in the CT–TMD interface generated a mutant (TMD-ct) that, while still sensitive to VRC01, was completely resistant to the trimer-specific bnAbs and also substantially sensitive to b6, 3791, and 17b (Fig. 4e). These results suggest that disrupting the packing interface between the TMD and CT can destabilize the Env ectodomain and shift it to an open conformation[10,40,41], further supporting our previous notion that the CT can modulate the antigenic structure of the Env trimer[5].

To further verify the impact of these mutations, we performed tier phenotyping[42] using seven HIV+ chronic serum samples with the 92UG037.8 (a tier 2 virus) wild-type and the TMD-ct mutant, along with the CT2-tmd and CT3-tmd mutants as representative "intermediate" phenotype viruses (Supplementary Table 3). As expected, the TMD-ct mutant became much more sensitive to the HIV+ chronic sera than the wild type, consistent with an open Env conformation and the tier-1 phenotype[42], while the two intermediate mutants were similar to the WT in the tier phenotype probably due to limited local changes near the trimer apex. We also tested additional bnAbs that target epitopes in CD4bs (binding site), V3-glycan, V1/V2-glycan, gp120/gp41 interface, and the MPER (Supplementary Table 4). While the wild-type and mutant viruses showed no major differences in sensitivity to most of these bnAbs, the mutants were ~10-times more sensitive to the MPER-directed antibody 4E10 than the wild-type virus, suggesting that the mutations in the CT and/or TM regions significantly alter the MPER structure.

**TMD dynamic parameters suggest pivotal motion of the TMD trimer.** Physical coupling between the CT and the apex of the ectodomain must be mediated via the TMD, but no obvious differences are observed between the NMR structures of the TMDs with and without the CT. We thus examined backbone

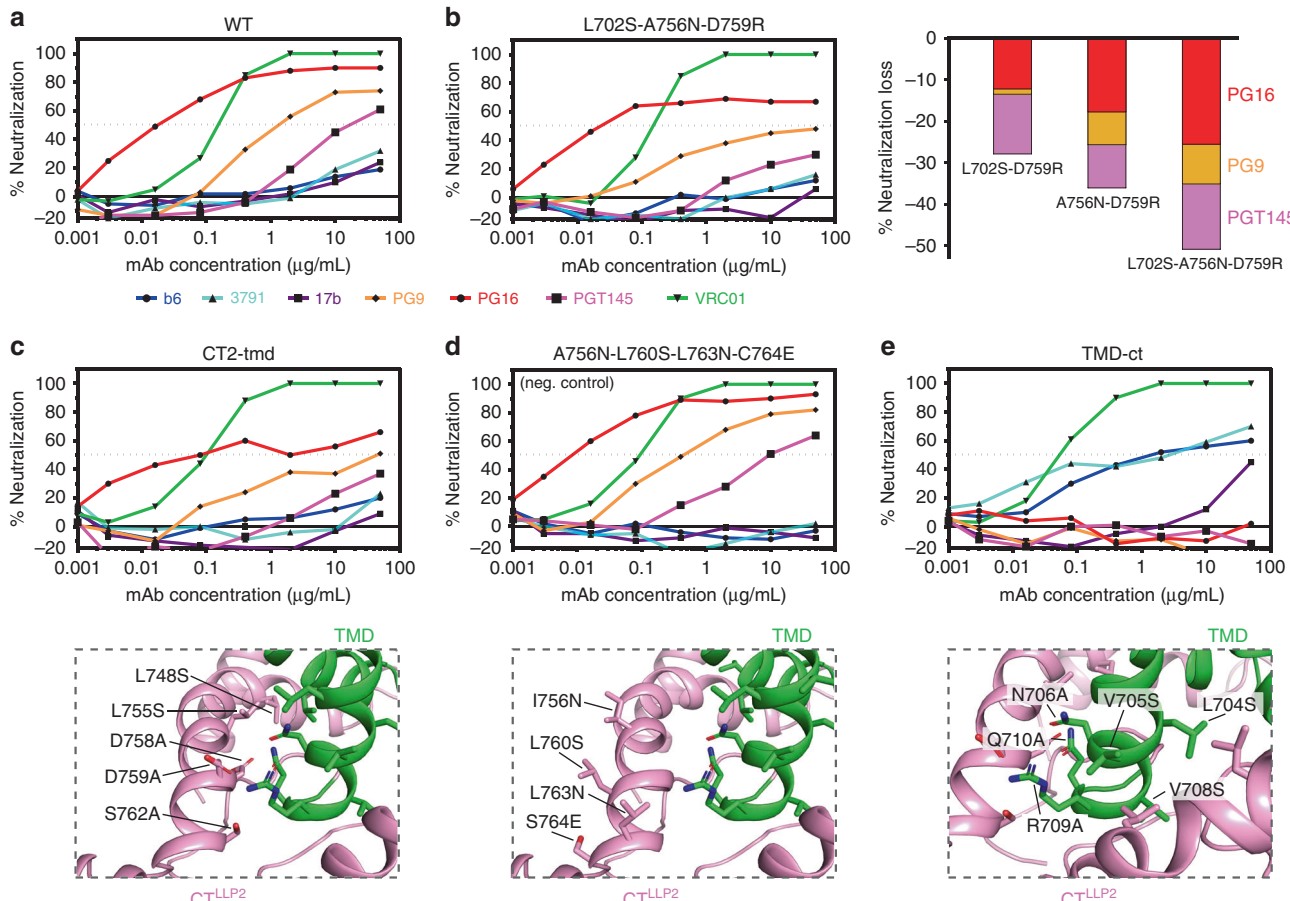

**Fig. 4 Effect of mutations in the CT on Env antibody sensitivity. a** Antibody neutralization of pseudovirus containing the 92UG037.8 Env determined for non-neutralizing antibodies, including b6 (CD4 binding site; blue), 3791 (V3; cyan), and 17b (CD4-induced; purple), and trimer-specific bnAbs, including PG9 (orange), PG16 (red), and PGT145 (magenta). The CD4 binding site bnAb VRC01, used as a control antibody, is shown in green. **b** Antibody neutralization of pseudovirus containing the L702S-A756N-D759R mutant. Right panel shows the relative decrease of sensitivity to the trimer-specific bnAbs of double and triple mutants bearing the D759R. **c** Antibody neutralization of pseudovirus (upper panel) containing a larger number of mutations in the CT (lower panel). **d** Same as (**c**) but for the mutant with mutations on the opposite side of the CT–TMD interface. **e** Same as (**c**) but for the mutant containing TMD mutations that break the hydrophobic and hydrophilic TMD–CT interactions. Source data are provided as a Source Data file.

dynamics across the TM in the MPER–TMD and TMD–CT$^{LLP2}$ constructs by measuring the product of NMR $^{15}$N $R_1$ and $R_2$ relaxation rates, $R_1R_2$, which is a probe for relative ms–µs motion associated with conformational exchange[43]. The MPER–TMD shows significantly higher dynamics in the MPER and C-terminal region of the TMD than the hydrophobic core of the TMD (residues 686–689), suggesting that both ends are unconstrained (Fig. 5a). We then locked the MPER by introducing the L660C and A667C mutations, which are positioned to form inter-chain disulfide in the MPER prefusion structure (Supplementary Fig. 17a). Locking the MPER resulted in substantially reduced motion not only in the MPER, as expected, but also in the C-terminal region of the TMD at the opposite side of the membrane; no change in the TMD hydrophobic core (Fig. 5a). Similar effects are observed in the TMD–CT$^{LLP2}$ where the $R_1R_2$ values for the TMD C-terminal region are less than that of the unlocked MPER–TMD (Fig. 5a). These data suggest that the three TM helices undergo a "scissor-like" movement around the hydrophobic core (or the hinge), at which the trimer remains tightly associated as shown by H–D exchange analysis (Supplementary Fig. 7 and ref. [26]), and that constraining either N- or C-terminal end of the TM helix with locked MPER or CT, respectively, can affect the opposite end of the TMD. Therefore, disruption or deletion of the CT baseplate can in principle destabilize the

MPER structure via the TMD (Fig. 5b). Since the MPER is a control relay that modulates the ectodomain conformation and antigenic properties[9], the CT-MPER coupling is also expected to shift conformational equilibria of the ectodomain and alter its antibody binding profile.

## Discussion

We have shown that the LLP2 segment of the Env CT forms amphipathic helices that wrap around the TMD, forming a support baseplate for the TMD and the rest of the Env. The other two LLP segments (LLP3 and LLP1) of the CT can also form amphipathic helices[19] and we suggest that they may fold around the LLP2 ring to further expand the baseplate (Fig. 5b). This model provides a structural explanation for our previous observations that progressive truncations of the CT incrementally reduced sensitivity of the Env ectodomain to PG16-like bnAbs, and that the effect reached maximum after the complete deletion of the LLP2[5], as progressive truncations of the CT would gradually weaken the base support for the TMD. Indeed, our data on the co-refolding of the TMD–KS and CT$^{LLP2}$ fragments show that the interaction between H1 and TMD is critical to the proper formation of the baseplate and possibly to the structural integrity of the entire Env. It is unclear whether the CT baseplate is

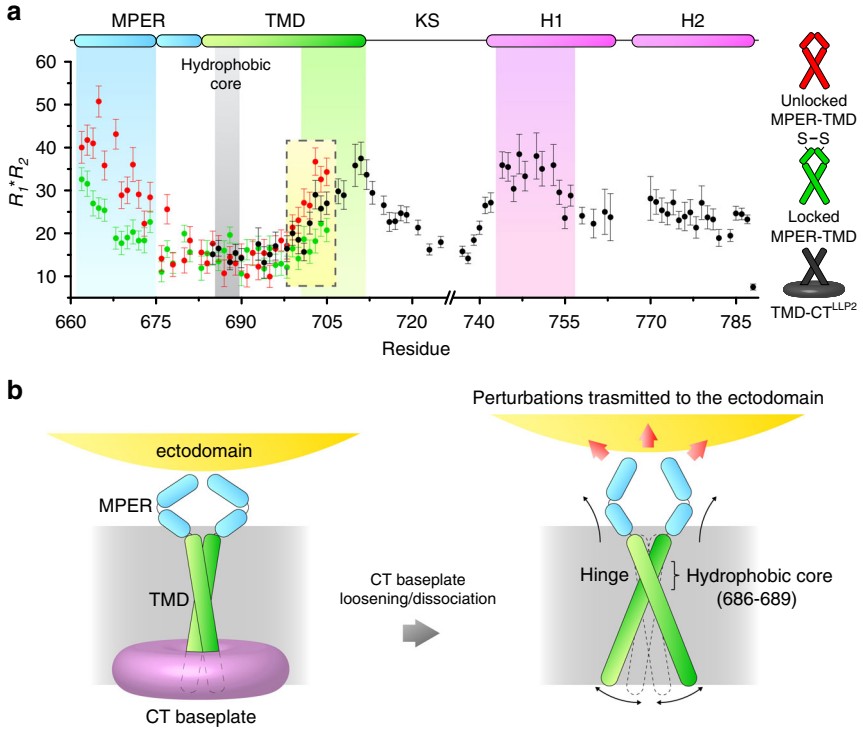

**Fig. 5 Conformational coupling between the Env CT and ectodomain. a** Backbone conformational dynamics of the unlocked MPER–TMD (red), locked MPER–TMD (green) and TMD–CT$^{LLP2}$ (black) probed by the product of $^{15}$N $R_1$ and $R_2$ relaxation rates, which reflects ms–μs motion. Error bars represent the uncertainty derived from $R_1$ and $R_2$ fitting errors. The regions characterized by higher $R_1R_2$, i.e., the MPER, the C-terminal part of the TMD, and its interacting CT region (H1), are highlighted in cyan, green, and purple, respectively. The position of the hydrophobic core (or the TM hinge) is shown in gray. The yellow box highlights the decrease of $R_1R_2$ in the C-terminal part of the TMD due to locking the MPER or CT presence. **b** Proposed mechanism of CT-ectodomain coupling mediated by the TMD. For simplicity, only two chains of the trimer are illustrated. Left: the CT baseplate confines the TMD motion, important for stabilizing the Env trimer in the prefusion state. Right: deletion of the CT baseplate or disruption of CT–TMD interactions allows greater TMD hinge motion, destabilizing the MPER, which in turn modulates the antigenic structure of the ectodomain. Source data are provided as a Source data file.

relevant to viral entry. The new structural information further consolidates the notion that conformational state of the membrane-related components of the Env can influence the antigenic structure of the ectodomain.

Rigorous understanding of the mode of coupling between the Env ectodomain and CT across the TMD would require visualization of the full-length Env at high resolution. Our NMR dynamics study of the MPER–TMD and TMD–CT$^{LLP2}$ fragments in bicelles provided compelling data suggesting physical coupling between the MPER and CT. The observation that locking the MPER with inter-chain disulfides reduced dynamics of the C-terminal end of the TMD ~50 residues away suggests that the proposed cross-membrane coupling of the Env is entirely possible by structural consideration. It was somewhat surprising that TMD C-terminal end showed slightly more dynamics in the TMD–CT$^{LLP2}$ than in the locked MPER–TMD, because we thought CT–TMD interaction may impose a more direct constraint on the TMD motion. Nevertheless, this result is not inconsistent with the proposed model in Fig. 5. Comparing to the unlocked MPER–TMD, the TMD–CT$^{LLP2}$ construct indeed showed reduced dynamics at the C-terminal end of the TMD, although the CT–TMD interaction still may not produce an effect as strong as covalently locking the MPER. The key message from the dynamics data is that the pivotal or the scissor-like motion of the TMD can be modulated by either MPER or CT.

To the best of our knowledge, a structural arrangement of a cytoplasmic baseplate supporting the ectodomain on the other side of membrane has never been observed in any known TM proteins including cellular receptors and viral fusion proteins.

Other lentiviral fusion proteins also have long CTs that are predicted to form multiple amphipathic helical segments, such as those from EIAV, MVV, and CAEV[17,18,44] (Supplementary Fig. 18). Whether these viral proteins adopt the unusual membrane-proximal fold seen for the HIV-1 Env CT remains to be investigated. Nevertheless, a trimeric baseplate of the CT together with the TMD appears to pose for specific interactions with the trimeric matrix (MA) protein[45–47], in the context of membrane.

In conclusion, our study provides a structural basis for how the CT of HIV-1 Env is physically coupled to its ectodomain through the TM and membrane-proximal regions and how the CT stabilizes the antigenic structure of the Env trimer on the opposite side of membrane. It is widely believed that the prefusion conformation of a functional HIV-1 Env on the surface of infectious virions is probably the most appropriate vaccine candidate for eliciting effective antibody responses, because it is recognized by most bnAbs and not by other tier-1 neutralizing antibodies[5,48,49]. Structural definition of the prefusion Env trimer at high resolution, which can provide a useful tool to facilitate immunogen selection and design, has limitations due to artificial modifications, extra ligands required for stability or the absence of membrane, compounded by observations with other spectroscopic approaches, such as single-molecule fluorescence resonance energy transfer (smFRET) and double electron–electron resonance (DEER) spectroscopy[50–52]. Our data presented here suggest new strategies to stabilize the native state of Env and help develop more effective Env trimer immunogens for clinical studies.

## Methods

**Protein expression and purification.** The gp41 fragment TMD–CT[LLP2] (residues 677–788) from a clade D HIV-1 isolate 92UG024.2 was synthesized by GenScript (Piscataway, NJ). The expression construct was created by fusing the TMD–CT[LLP2] to the C-terminus of the His[9]-TrpLE expression sequence in pMM-LR6 vector with an added methionine in-between for subsequent cleavage during protein purification, as previously described[8,9,23]. Mutants and additional constructs (e.g., TMD–KS and CT[LLP2], Supplementary Tables 5 and 6) were generated by standard PCR protocols and confirmed by DNA sequencing.

Each protein construct was expressed by growing transformed *E. coli* strain BL21 (DE3) cells in LB or M9 minimal media (when isotopic labeling was required). Cultures were grown at 37 °C until they reached an optical density of ~0.6 and were then cooled to 20 °C before induction with 0.1 mM isopropyl β-D-thiogalatopyranoside (IPTG). Protein was expressed at 20 °C for ~18–24 h.

After growth, cells were harvested, suspended in a lysis buffer (50 mM Tris, pH 8.0, and 200 mM NaCl), and lysed by sonication. Inclusion bodies were separated by centrifugation at $25,400 \times g$ and suspended in a denaturing buffer (1% Triton X-100, 6 M guanidine hydrochloride, 50 mM Tris, pH 8.0, and 200 mM NaCl). Inclusion bodies were homogenized using a glass tissue grinder, dissolved and centrifuged at $25,400 \times g$. The fusion protein was bound to nickel affinity resin (Sigma-Aldrich), washed with 8 M Urea and dH$_2$O, and finally eluted with 90% formic acid (FA). The gp41 construct was cleaved from the TrpLE by hydrolyzing the peptide bond at the C-terminus of the methionine using cyanogen bromide (CNBr) (~0.1 g/mL) in 90% FA for 1 h. The reaction mixture was dialyzed (MWCO 3.5 kDa) to remove the excess of CNBr and FA and then lyophilized. The protein powder was dissolved in 90% FA and purified by reverse-phase high-performance liquid chromatography (RP-HPLC) in a Zorbax SB-C3 column (Agilent Technologies, Santa Clara, CA) with a gradient from 95% dH$_2$O, 5% isopropanol (IPA), 0.1% trifluoroacetic acid (TFA) (buffer A) to 75% IPA, 25% acetonitrile, 0.1% TFA (buffer B) (Supplementary Fig. 2a-b). Fractions containing the pure gp41 construct (confirmed by SDS-PAGE analysis) were collected and lyophilized.

**Protein reconstitution.** The lyophilized gp41 construct was dissolved in 1,1,1,3,3,3-hexafluoro-2-propanol (HFIP) and mixed with 9 mg of 1,2-Dimyristoyl-sn-glycero-3-phosphocholine (DMPC) (from Avanti Polar Lipids, Alabaster, AL or from FB Reagents, Boston, MA, if deuterated) and 27 mg of 1,2-Dihexanoyl-sn-glycero-3-phosphocholine (DHPC) (from Avanti Polar Lipids Alabaster, AL or from FB Reagents Boston, MA, if deuterated). The mixture was dried under a nitrogen stream until a thin film was achieved and lyophilized overnight to completely remove any trace of residual organic solvent. The thin film was then dissolved in 3 mL of 8 M Urea and dialyzed (MWCO 3.5 kDa) against 40 mM MES buffer, pH 6.7 to remove the denaturant. During and after the dialysis, additional DHPC was added to make up for the DHPC lost during the dialysis, adjusting the DMPC:DHPC ratio ($q$) to ~0.5. The reconstituted protein was concentrated using a Centricon (EMD Millipore, Billerica, MA) (MWCO 3.5 kDa) to ~300–350 μL. The buffer of the final NMR sample contained 40 mM DMPC, 80 mM DHPC, 40 mM MES, pH 6.7, 1% NaN$_3$, and 10% (v/v) D$_2$O (for the NMR lock). Finally, the bicelle $q$ of the NMR sample was quantified by signal integration of the DMPC and DHPC methyl peaks in the 1D $^1$H NMR spectrum and adjusted to exactly 0.5.

**Determination of the protein oligomeric state.** To determine the oligomeric state of the bicelle-reconstituted TMD–CT[LLP2], TMD–KS, and CT[LLP2], standard SDS-PAGE analysis was initially used. The proteins were reconstituted in DMPC/DHPC bicelles at $q = 0.5$, mixed with an SDS sample buffer (Invitrogen) and boiled for 5 min, followed by SDS-PAGE (12% Bis-Tris protein gel) at 200 V for 30 min and Commassie blue staining. The TMD–KS migrated at the apparent MW of ~17 kDa (theoretical MW is 5.8 kDa), indicating that it had remained trimeric despite SDS denaturation as previously shown[8,9] (Supplementary Fig. 9b, lane 2 of the gel). The TMD–CT[LLP2] and CT[LLP2], instead, migrated at the apparent MW of ~12 and ~7 kDa (theoretical MW are 11.8 and 6.1 kDa, respectively), indicating that both had become monomeric upon SDS denaturation (Supplementary Fig. 1b, lane 1 of the gel, and Supplementary Fig. 9, lanes 1 and 2 of the gels, respectively). Therefore, their native oligomeric states were quantified using the non-denaturing method known as OG-label[23,27].

In the OG-label method, each protomer of the oligomer to be studied is non-covalently labeled with a soluble cross-linkable protein (SCP), so that the latter can be cross-linked using Lomant's reagents to read out the sample oligomeric state. The small Ig-fold protein named GB1 (MW = 8.4 kDa) has been proven to serve as the SCP very effectively. A TriNTA molecule is linked via PEG-2-SMCC (succinimidyl 4-(N-maleimidomethyl)cyclohexane-1-carboxylate) to the N-terminus of GB1, while a His$_6$-tag is added to the C-terminus of the oligomeric protein, so that the TriNTA-GB1 conjugate can strongly attach to the protomer (the binding affinity of TriNTA to His$_6$-tag is $20 \pm 10$ nM). The GB1s are then cross-linked to report the oligomeric state of the protein, as the local concentration of stoichiometric amount of GB1 to the oligomer allows for more efficient cross-linking than for the free GB1 in solution. Finally, the cross-linked GB1s are released from the oligomer by addition of EDTA and analyzed by SDS-PAGE.

To implement the OG-label method for the TMD–CT[LLP2] and CT[LLP2], a His$_6$-tag was added at the C-terminus of the two proteins. Three samples were prepared

containing either (i) TMD–CT[LLP2], (ii) CT[LLP2] or (iii) CT[LLP2] in presence of TMD–KS in ~1:1 ratio. The His$_6$-tagged proteins were expressed, purified and reconstituted in bicelles ($q = 0.5$) as previously described, except the sample buffer was a phosphate buffer (pH 7.4) for better cross-linking efficiency. To prevent unwanted cross-linking between the membrane protein and GB1, all the free primary amines of TMD–CT[LLP2], CT[LLP2], or TMD–KS were blocked by reacting with 100-fold molar excess of Sulfo-NHS acetate (Thermo Fisher Scientific) at room temperature for 1.5 h. Excess Sulfo-NHS acetate was removed by dialysis while tightly controlling the bicelle $q$. After dialysis, the samples were concentrated to 40 μM and mixed with 60 μM TriNTA-GB1 to ensure that all the His$_6$-tags were saturated with TriNTA-GB1. The mixtures were then incubated at room temperature with various concentration of BS3(PEG9) (0.3, 0.9, 3.0, and 3.0 mM, corresponding to lane 4, 5, 6, and 7 in the gels shown in Supplementary Figs. 1b and 9, respectively) for 30 min, followed by a second incubation at room temperature with various concentration of glutaraldehyde (0.6, 0.6, 0.6, and 1.8 mM, corresponding to lane 4, 5, 6, and 7 in the gels shown in Supplementary Figs. 1b and 9, respectively) for 3 min. The cross-linking reactions were quenched with 20 mM Tris buffer (pH 7.5) upon incubation at room temperature for 15 min. As negative control, 0.9 mM BS3(PEG9) and 0.6 mM glutaraldehyde were sequentially added to 60 μM TriNTA-GB1 in the absence of the His$_6$-tagged samples (lane 3 in the gels shown in Supplementary Figs. 1b and 9), indicating that the TriNTA-GB1 alone remains mostly monomeric and only partially dimerizes in the conditions used. The cross-linked GB1s were then released from the samples by adding 50 mM EDTA and examined by SDS-PAGE using 12% Bis-Tris protein gels (Thermo Fisher Scientific) (Supplementary Figs. 1b and 9).

**NMR data acquisition and processing.** The NMR experiments were performed at a) 14.1 T on a Bruker Avance III HD spectrometer operating at 600.13 MHz $^1$H, 150.90 MHz $^{13}$C, and 60.81 MHz $^{15}$N frequencies; (b) 17.6 T on a Bruker Avance III spectrometer operating at 749.66 MHz $^1$H, 188.50 MHz $^{13}$C, and 75.96 MHz $^{15}$N frequencies; (c) 18.8 T on a Bruker Avance III spectrometer operating at 800.28 MHz $^1$H, 201.23 MHz $^{13}$C, and 81.09 MHz $^{15}$N frequencies; (d) 21.1 T on a Bruker Avance III spectrometer operating at 900.17 MHz $^1$H, 226.35 MHz $^{13}$C, and 91.21 MHz $^{15}$N frequencies. All the spectrometers were equipped with a cryogenic probe. All the measurements were performed at 308 K if not stated otherwise. The most relevant acquisition parameters of the experiments are reported in Supplementary Table 7.

The NMR data sets were processed with *nmrPipe*[53] and the resulting NMR spectra were analyzed with *Sparky* (T. D. Goddard and D. G. Kneller, SPARKY 3, University of California, San Francisco) and *XEASY*[54]. Peak intensities were measured at peak local maxima using quadratic interpolation to identify peak centers. *Origin* (OriginLab, Northampton, MA) was used to fit the experimental data. For comparison purpose, the chemical shift assignments of the TMD and MPER–TMD were taken from the Biological Magnetic Resonance Bank (BMRB), entries 30090 and 30503, respectively[8,9]; the TMD and MPER–TMD average structures were taken from the Protein Data Bank (PDB), entries 5JYN and 6E8W, respectively[8,9].

**NMR resonance and NOE assignment.** Sequence specific assignment of TMD–CT[LLP2] backbone chemical shifts (BMRB accession code 30678) was accomplished using a set of TROSY-enhanced triple resonance experiments (HNCA, HN(CO)CA, HN(CA)CO, HNCO and HNCACB)[55,56], recorded on a ($^{15}$N, $^{13}$C, 85% $^2$H)-labeled sample. In addition, an ultra-high-resolution 3D $^{15}$N-edited NOESY-TROSY-HSQC ($\tau_{mix} = 180$ ms) spectrum of a ($^{15}$N, $^2$H)-labeled sample resulted extremely useful in completing the backbone resonance assignment. Protein aliphatic and aromatic resonances were assigned using a combination of 2D $^{13}$C HSQC, 3D $^{15}$N-edited NOESY-TROSY-HSQC ($\tau_{mix} = 80$ ms), and $^{13}$C-edited NOESY-HSQC ($\tau_{mix} = 150$ ms). These NOESY experiments were performed on a ($^{15}$N, $^{13}$C)-labeled sample reconstituted in bicelles with deuterated DMPC and DHPC acyl chains.

NOE-derived intra-chain distance restraints for the TMD–CT[LLP2] were obtained from the 3D $^{15}$N-edited and $^{13}$C-edited NOESY spectra used for aliphatic and aromatic resonance assignment above. Assigning inter-chain distance restraints, however, faced the challenge of measuring NOEs between structurally equivalent protomers having the same chemical shifts. To overcome this problem, we used a mixed sample in which half of the protomers were $^{15}$N, $^2$H-labeled and the other half $^{13}$C-labeled. Recording a 3D $^{15}$N-edited NOESY-TROSY-HSQC ($\tau_{mix} = 200$ ms) on this sample allowed identification of NOEs exclusively between the $^{15}$N-attached protons of one protomer and the aliphatic protons of the neighboring one. The identified NOEs were unambiguously confirmed by performing the 3D $J_{CH}$-modulated NOE experiment[9,23], in which two interleaved NOESY spectra were recorded with varying the $J_{CH}$ evolution ($J_{CH} = 0$ ms and $J_{CH} = 8$ ms) before the NOE mixing. Subtraction of the two spectra allowed selection of only the inter-chain NOEs. Finally, a control sample containing only ($^{15}$N, $^2$H)-labeled protein was used to perform an identical 3D $^{15}$N-edited NOESY-TROSY-HSQC experiment as done for the mixed sample above (note: the $^{15}$N, $^2$H-labeled TMD–CT[LLP2] in the mixed and control samples were from the same protein expression batch). Comparison of the mixed and control spectra provided another confirmation that the detected inter-chain NOEs were due to the mixing of protomers with different labeling schemes and not to residual protonation. This

assignment strategy is summarized in Supplementary Fig. 4, where several examples of unambiguously assigned inter-chain NOEs are shown.

**NMR-based membrane partition analyses.** The membrane partition of the TMD–CT[LLP2] was determined using the paramagnetic probe titration (PPT) method[23,25]. As previously shown, DMPC/DHPC bicelle with sufficiently large $q$ ($\geq 0.5$) allows direct use of measurable paramagnetic relaxation enhancement (PRE) to probe residue-specific immersion depth of the protein in the bilayer region of the bicelle. Two PPT analyses were performed: titrating the bicelle-reconstituted TMD–CT[LLP2] with (1) the soluble paramagnetic agent Gd-DOTA, and (2) the lipophilic paramagnetic agent 16-Doxyl-stearic acid (16-DSA). The titrants were taken from concentrated stock solutions (600 mM Gd-DOTA and 24 mM 16-DSA) in the same buffer as that of the protein sample and were added in small aliquots (few μL per step) to minimize sample dilution. The PRE increase was monitored by recording a 2D $^1$H–$^{15}$N TROSY-HSQC spectrum at each of the titrant concentrations: 0 (reference), 2.0, 4.0, 6.0, 8.0, 10.0, 15.0, and 20.0 mM for Gd-DOTA; 0 (reference), 0.6, 1.2, 1.8, 2.4, 3.0, 3.6, and 4.2 mM for 16-DSA. The residue-specific PRE[amp], which is the amplitude of the PRE experienced by an amide proton in the protein, was determined by fitting the peak intensity decay as a function of [paramagnetic probe] to the following exponential decay equation:

$$\frac{I}{I_0} = 1 - \text{PRE}_{\text{amp}}\left(1 - e^{-\frac{[\text{paramagnetic probe}]}{\tau}}\right) \quad (1)$$

where $I$ and $I_0$ are the peak intensities in the presence and absence of the paramagnetic probe, respectively, [paramagnetic probe] is the concentration of the paramagnetic agent (Gd-DOTA or 16-DSA), $\tau$ is the decay constant and PRE[amp] is the PRE amplitude. The residue-specific PRE[amp] (Supplementary Table 8) were then used to determine the membrane partition of the protein ($r_Z$) by the sigmoidal fitting method[23,25], in which the position of the TMD–CT[LLP2] trimer along the bilayer normal was fitted to satisfy Eq. (2):

$$\text{PRE}_{\text{amp}} = \text{PRE}_{\text{amp}}^{\min} + \frac{(\text{PRE}_{\text{amp}}^{\max} - \text{PRE}_{\text{amp}}^{\min})}{1 + e^{(r_Z^I - |r_Z|)/\text{SLOPE}}} \quad (2)$$

where $\text{PRE}_{\text{amp}}^{\min}$ and $\text{PRE}_{\text{amp}}^{\max}$ are the limits within which PRE[amp] can vary for a particular protein system, $r_Z^I$ is the inflection point (the distance from the bilayer center at which PRE[amp] is halfway between $\text{PRE}_{\text{amp}}^{\min}$ and $\text{PRE}_{\text{amp}}^{\max}$), and SLOPE is a parameter which reports the steepness of the curve at the inflection point.

**Measurement of inter-chain PREs.** Inter-chain PREs were measured for cross validation of the NOE-derived TMD–CT[LLP2] structure (Supplementary Fig. 6). Three mixed samples were prepared by mixing, at ~1:1 molar ratio, the ($^{15}$N, 85% $^2$H)-labeled TMD–CT[LLP2] with one of the following unlabeled TMD–CT[LLP2] construct containing a specific Cys mutation/addition: (a) G738C; (b) S764C (reintroducing the native cysteine); (c) C789 (Cys added to the C-terminus). The mutant proteins were prepared as in the "Protein expression and purification" section above. After reconstitution in bicelles under reducing conditions, DTT was removed from the samples by dialysis at low pH (6.2). The pH was then rapidly raised to 7.4 and 100 mM MTSL (in DMSO) was added to a final ratio of 10:1 (MTSL to Cys-mutant TMD–CT[LLP2]) and allowed to react at room temperature overnight. Excess MTSL was removed by extensive dialysis (pH 6.7). The samples were then concentrated to 360 μL for NMR measurements. During the entire sample preparation, MTSL-containing solutions were shielded from light. As a negative control, a sample containing only ($^{15}$N, 85% $^2$H)-labeled TMD–CT[LLP2] (without Cys) was prepared using the same MTSL-labeling procedure used for the mixed samples, to ensure that free MTSL removal with our protocol was complete (Supplementary Fig. 6d). To quantify the PREs, defined as the ratio of the peak intensities in the paramagnetic ($I$) and diamagnetic state ($I_0$), a 2D $^1$H–$^{15}$N TROSY-HSQC spectrum was recorded before and after reducing the spin-label by addition of 20 mM sodium ascorbate (pH 6.7).

The same protocol was used to study the interaction between the TMD and the CT (Figs. 2 and 3). Three mixed samples were prepared by mixing: (1) ($^{15}$N, 85% $^2$H)-labeled TMD–KS with unlabeled CT[LLP2] C789 (~1:1); (2) ($^{15}$N, 85% $^2$H)-labeled CT[LLP2] C789 and unlabeled TMD–KS (~1:2:3); (3) ($^{15}$N, 85% $^2$H)-labeled CT2-tmd (L748S-L755S-D758A-D759A-S762A) with unlabeled CT2-tmd S764C (~1:1).

**Measurement of H–D exchange.** Solvent accessibility of the TMD–CT[LLP2] was examined by performing an H–D exchange experiment at 303 K (Supplementary Fig. 7). The TMD–CT[LLP2], reconstituted in protonated solvent (pH 6.0), was flash-frozen in liquid nitrogen and thoroughly lyophilized. The dried sample was then dissolved in 360 μL of 99.9% D$_2$O (pD ~ 6.4). The progress of the H–D exchange was monitored by measuring a 2D $^1$H–$^{15}$N TROSY-HSQC spectrum at uniform time intervals of ~3 h up to ~3.25 days. The residue-specific exchange constant, $k_{\text{ex}}$ ($=1/\tau_{\text{ex}}$) (Supplementary Table 9), was determined by fitting the fractional peak intensity vs. time to the following exponential decay equation:

$$I_{(t)} \propto e^{-\frac{t}{\tau_{\text{ex}}}} \quad (3)$$

where $I$ is the peak intensity, $t$ is the time passed from the beginning of the

exchange, and $\tau_{\text{ex}}$ is the time constant of the decay. Finally, $k_{\text{ex}}$ values were divided in four different exchange regimes defined as: very fast ($\tau_{\text{ex}} < 1$ h), fast (1 h $\leq \tau_{\text{ex}} < 3$ h), slow (3 h $\leq \tau_{\text{ex}} < 1$ day), and very slow ($\tau_{\text{ex}} \geq 1$ day).

**Measurement of NMR relaxation rates.** Backbone $^{15}$N dynamics of the bicelle-reconstituted TMD–CT[LLP2] (14.1 T, 303 K) was investigated by measuring $^{15}$N $R_1$ and $R_2$ relaxation rates using the TROSY version of the standard experiments[57] (Supplementary Fig. 17b). For $^{15}$N $R_1$, 8 experiments were acquired with the following relaxation delays: 10, 50, 100, 200, 300, 600, 800, and 1000 ms. For $^{15}$N $R_2$, 8 experiments were acquired with the following relaxation delays: 6.4, 10, 20, 30, 40, 50, 64, and 80 ms. The $^{15}$N $R_1$ and $R_2$ values were determined by fitting the peak intensity vs. relaxation delay to the exponential decays:

$$I_{(t)} \propto e^{-R_1 t} \quad (4)$$

$$I_{(t)} \propto e^{-R_2 t} \quad (5)$$

where $I$ is the peak intensity at a given relaxation delay, $t$ is the relaxation delay, and $R_1$ and $R_2$ are the relaxation rates.

The same type of analysis was performed also on the unlocked and locked MPER–TMD. The unlocked MPER–TMD was prepared by reconstituting in bicelles a MPER–TMD mutant (L660C, A667C) under reducing condition (10 mM DTT) following the same protocol used for the wild-type MPER–TMD[9]. The production of the mutant MPER–TMD followed the same protocol used for the wild-type MPER–TMD[9]. Briefly, a fragment of HIV-1 gp41 (clade D, isolate 92UG024.2) containing residues 660–710 was expressed in *E. coli* strain BL21 (DE3) cells as a trpLE fusion and purified as in the "Protein expression and purification" section above. The locked MPER–TMD was subsequentially obtained upon complete removal of the DTT by dialysis and gradual addition of Glutathione (ox) to the final concentration of 10 mM. The complete locking of the MPER was confirmed by Urea-PAGE and mass spectrometry analyses (Supplementary Fig. 17a).

**Plane restraints.** The PPT method[23,25] was used to derive a set of plane restraints to aid structure calculation. Provided that the bicelle is sufficiently large ($q \geq 0.5$), the PPT method allows to determine the projection of each amide proton ($r_Z$) along the protein C$_3$ symmetry axis, which is also parallel to the bicelle normal and aligned to the Cartesian Z axis for convenience. Therefore, $r_Z$ can be assigned as residue-specific plane restraint if the PRE[amp] values are calibrated against a known structure with known membrane partition. Since the TMD–CT[LLP2] and the TMD show remarkable chemical shift and structure similarity for residues 677–710 (Supplementary Figs. 1c, 3a, and 8), this region was used to calibrate the PRE[amp] values of the TMD–CT[LLP2] (from both Gd-DOTA and 16-DSA) against the previously established structure and membrane partition of the TMD[8,26] (Supplementary Fig. 5e). Specifically, we generated the PRE[amp] vs. $r_Z$ plot for residues 679–710 using the residue-specific PRE[amp] from the TMD–CT[LLP2] and the known $r_Z$ from the known TMD structure. The data were then fitted to the sigmoidal function (Eq. (2)) to yield parameterized Eq. 2 for Gd-DOTA and 16-DSA, which were then used to determine $r_Z$ for the other residues of the TMD–CT[LLP2] not used for the calibration (711–788). Out of the calculated $r_Z$, only those in the sensitive region of the sigmoidal curves were retained. Finally, $r_Z$ derived from Gd-DOTA and 16-DSA data sets were averaged and merged into one single data set, yielding the final set of plane restraints. The plane restraints used for the TMD–CT[LLP2] structure calculation included $r_Z$ from the published TMD structure (residues 679–710) and the newly calculated $r_Z$ for residues 711–788, with an uncertainty of ±1 Å.

Using the same procedure, plane restraints were also assigned for the MPER and included in the MPER–TMD–CT[LLP2] model calculation. In this case, the previously published PRE[amp] values of the MPER–TMD (from Gd-DOTA titration)[9] were calibrated against residues 695–708 of the TMD (Supplementary Fig. 5g) and used to derive plane restraints for residues 660–694.

The data used to generate the plane restraints for the TMD–CT[LLP2] and MPER–TMD–CT[LLP2] (Supplementary Table 10) are summarized in Supplementary Fig. 5.

**Structure calculation.** NMR structures were calculated using *XPLOR-NIH*[58]. Since the TMD–CT[LLP2] (677–788) is an extension of the previously studied TMD (677–716) and both constructs exhibit almost identical chemical shift for residues 677–710 (Supplementary Figs. 1c, 3a, and 8), we used the structure of the TMD[8] as a starting point for the new calculation. To implement this strategy, we first performed *TALOS+*[59] analysis using the assigned backbone chemical shift values ($^{15}$N, $^{13}$C$^\alpha$, $^{13}$C$^\beta$, and $^{13}$C') and then used the "GOOD" dihedral angles generated by TALOS+ to build a secondary structural model of the TMD–CT[LLP2]. Second, we assembled a trimer using the previously assigned NOE restraints for residues 677–702 (from PDB accession code 5JYN). Finally, we applied the newly assigned inter-chain restraints (most of them from residues 704–785) to complete the trimer of the TMD–CT[LLP2]. For each inter-chain restraint between two adjacent protomers, three identical distance restraints were assigned respectively to all pairs of neighboring protomers to satisfy the condition of C$_3$ symmetry. The assembled trimer was then refined against the complete set of NOE restraints (intra- and inter-chain) and dihedral angles using a simulated annealing (SA) protocol in

which the temperature was lowered from 1000 to 200 K in steps of 50 K. The NOE restraints were enforced by flat-well harmonic potentials, with the force constant ramped from 2 to 40 kcal/mol Å$^{-2}$ during annealing. Backbone dihedral angle restraints were enforced by flat-well (± the corresponding uncertainties from TALOS+) harmonic potential with force constant ramped from 10 to 1000 kcal/mol rad$^{-2}$. The plane restraints were fixed in space and enforced by flat-well (±1 Å) harmonic potentials, with force constant ramped from 1 to 5 kcal/mol Å$^{-2}$ during annealing. A total of 150 structures were calculated, and the 15 lowest energy structures were selected as the final structural ensemble (PDB accession code 6UJU) (Supplementary Fig. 19b and Supplementary Table 11).

The model of the MPER–TMD–CT$^{LLP2}$ was generated in a similar manner. The matching resonances of the TM core (residues 685–700) in the MPER–TMD (660–710), TMD (677–716), and TMD–CT$^{LLP2}$ (677–788) (Supplementary Figs. 1c, 3a, and 8) allowed to merge the three structures at the TM core to generate a MPER–TMD–CT$^{LLP2}$ starting model for further refinement. NOEs and backbone dihedral angle restraints for the MPER and TMD (residues 660–693 and 694–710, respectively) were taken from previously published studies[8,9], while those for the CT$^{LLP2}$ (711–788) were taken from the current study. Plane restraints from the three regions were also applied. The model was refined using identical parameters and potentials as those used for the TMD–CT$^{LLP2}$ (see above). A total of 150 structures were calculated, and the 15 lowest energy structures were selected as the final structural ensemble (PDB accession code 6UJV) (Supplementary Fig. 19c and Supplementary Table 12).

**Env mutant constructs and monoclonal antibodies.** Full-length Env mutants were generated using the 92UG037.8 gp160[5] as a template by QuikChange Site-Directed Mutagenesis (Agilent Technologies). All constructs were confirmed by restriction digestion and DNA sequencing. Anti-HIV-1 Env monoclonal antibodies and their Fab fragments were produced as previously described[5,6]. Briefly, the intact antibodies or Fab fragments were expressed in 293T cells either by transient transfection or using selected stably transfected clones, or from hybridomas or CHO cells. The antibodies were purified by affinity chromatography using Gamma bind plus resin (GE Healthcare, Pittsburgh, PA), followed by gel-filtration chromatography. Expression constructs of antibodies PG9, PG16, and PGT145 were generated using synthetic genes made by GeneArt Gene Synthesis (Life Technologies, Carlsbad, CA) or GenScript (Supplementary Table 13). The VRC01 expression constructs were kindly provided by John Mascola (VRC, NIH); the CHO stable line expressing antibody b6 by Dennis Burton (Scripps); 17b hybridoma by James Robinson (Tulane University); 3791 hybridoma by Susan Zolla-Pazner (New York University).

**Production of pseudoviruses containing mutant Envs.** Preparation of HIV-1 Env pseudoviruses of CT mutants, and titration of pseudovirus stocks to determine the 50% tissue culture infectious dose per mL (TCID$_{50}$/mL) were performed as previously described[8,60]. Briefly, each pseudovirus was prepared by transfecting 293T/17 cells (ATCC, Manassas, VA) with 4 µg of Env expression plasmid and 8 µg of an Env-deficient HIV-1 backbone vector (pSG3ΔEnv). Pseudovirus-containing culture supernatant was harvested 24 h after transfection, filtered (0.45 µm), and stored at −80 °C. To determine TCID$_{50}$/mL, a 5-fold serial dilution of virus stock was performed in quadruplicate wells and incubated with TZM.bl cells (NIH AIDS reagent program) in growth media containing DEAE-dextran (Sigma-Aldrich) at a final concentration of 11 µg/mL. After 48 h, the cells were measured for luciferase reporter gene expression, indicating the ability of the pseudovirus to infect cells. TCID$_{50}$/mL was calculated using an Excel macro made available on the Las Alamos National Laboratories website (www.hiv.lanl.gov).

**HIV-1 p24 antigen ELISA assay and western blot.** Viral stocks were boiled in a buffer containing 0.5% Triton X-100 for 60 min and analyzed for p24 antigen using a HIV-1 p24 antigen ELISA 2.0 kit (ZeptoMetrix Corporation, Buffalo, New York)[8]. Virus lysates were made by directly mixing p24-normalized virus stocks with Laemmli Sample Buffer (Bio-Rad, Hercules, CA) and boiling for 10 min. Lysates of cells expressing Env or its mutants were prepared by resuspending the cells in PBS (Phosphate-Buffered Saline) at a density of $2.0 \times 10^6$ cells/mL, followed by treatment with the Sample Buffer and boiling for 10 min. Western blot was performed following our published protocol[8]. Briefly, Env samples were resolved in 4–15% Mini-Protean TGX gel (Bio-Rad) and transferred onto PVDF membranes (Millipore, Billerica, MA) by an Iblot2 (Life Technologies). Membranes were blocked with 5% skimmed milk in PBS for 1 h and incubated with anti-V3 loop antibody 3791 for another hour at room temperature. Alkaline phosphatase conjugated anti-human Fab IgG (1:5000) (Sigma-Aldrich, LOT number: A8542) was used as a secondary antibody. Env proteins were visualized using one-step NBT/BCIP substrates (Thermo Scientific, Cambridge, MA).

**Flow cytometry.** 293T cells were transiently transfected with 2 µg of the 92UG037.8 gp160 expression construct or its CT mutants in six-well plates. Flow cytometry was carried out as previously described[5,8]. Briefly, Env-expressing cells were detached from plates using PBS and washed with ice-cold PBS containing 1% BSA. Hundred and six cells were incubated for 30–40 min on ice with either VRC01 Fab, 2G12 Fab, or PG16 IgG at concentrations of 10 µg/ml in PBS

containing 1% BSA. The cells were then washed twice with PBS containing 1% BSA and stained with R-Phycoerythrin AffiniPure F(ab′)2 fragment goat anti-human IgG, F(ab′)2 Fragment specific secondary antibody (Jackson ImmunoResearch Laboratories, West Grove, PA) at 5 µg/ml. All the fluorescently labeled cells were washed twice with PBS containing 1% BSA and analyzed immediately using a BD FACScanto II instrument and program FACSDIVA (BD Biosciences, San Jose, CA). All data were analyzed by *FlowJo* (FlowJo, LLC, Ashland, OR). The flow cytometry gating strategy is shown in Supplementary Fig. 20.

**Cell–cell fusion assay.** Cell–cell fusion assay was based on the α-complementation of *E. coli* β-galactosidase[8]. Briefly, 293T cells were cotransfected with either HIV-1 Env and the α fragment of β-galactosidase or CD4, CCR5 and the ω fragment of β-galactosidase. Env-expressing cells ($2.0 \times 10^6$ cells/mL) were mixed with CD4- and CCR5-expressing cells ($2.0 \times 10^6$ cells/mL). Cell–cell fusion was allowed to proceed at 37 °C for 2 h. Cell–cell fusion activity was quantified using a chemiluminescent assay system, Gal-Screen (Applied Biosystems, Foster City, CA).

**Viral infectivity and antibody neutralization assays.** Viral infectivity of HIV-1 92UG037.8 Env and the CT mutants was measured by infecting TZM.bl cells with p24-normalized pseudovirus in growth media containing DEAE-dextran (11 µg/mL). Forty eight hours post-infection, luciferase activity of the reporter gene was quantified[31]. Likewise, neutralizing IC50 and IC80 titers of monoclonal antibodies were determined also by the luciferase-based virus neutralization assay in TZM.bl cells, which measures the reduction in luciferase reporter gene expression in TZM-bl cells following a single round of virus infection (Supplementary Table 2)[31]. Briefly, 5-fold serial dilutions of antibody samples were performed in duplicate (96-well flat-bottom plate) in 10% DMEM growth medium (100 µL/well). Virus was added to each well in a volume of 50 µL, and the plates were incubated for 1 h at 37 °C. TZM.bl cells were then added ($1.0 \times 10^4$/well in 100 µL volume) in 10% DMEM growth medium containing DEAE-Dextran. Following a 48 h incubation, luminescence was measured with Bright-Glo luciferase reagent (Promega, Madison, WI). The IC50 and IC80 titers were calculated as the antibody dilution that resulted in a 50% or 80% reduction in relative luminescence units (RLU), respectively, compared with the virus control wells after the subtraction of cell control RLU. Maximum percent inhibition (MPI) indicates the highest percent inhibition of virus infection observed with the tested concentrations of each antibody. Murine leukemia virus (MuLV) was used as a negative control virus for all assays. Antibodies used in this assay include IgG of b6, 3791, 17b, PG9, PG16, PGT145, VRC01, 3BNC117, PGT121, 10-1074, PGDM1400, PGT151, 8ANC194, and 4E10. The tier phenotyping assay used the following HIV+ chronic serum samples: HIV-018, HIV-019, HIV-021, HIV-023, HIV-024, HIV-025, and HIV-026, which are all chronic clade B HIV+ serum. The assay was described previously[42] and contained in the same protocol described above. The IC50 and IC80 titers were calculated as the serum dilution that caused a 50 and 80% reduction in relative RLU compared with the level in the virus control wells after subtraction of cell control RLU.

**Reporting summary.** Further information on research design is available in the Nature Research Reporting Summary linked to this article.

## Data availability
The source data underlying Figs. 2–5 and Supplementary Figs. 1, 2, 5, 6, 8, 9, 11, 13-17 are provided as a Source Data file. The atomic structure coordinate and structural constraints have been deposited in the Protein Data Bank (PDB), accession numbers 6UJU and 6UJV. The chemical shift values have been deposited in the Biological Magnetic Resonance Data Bank (BMRB), accession number 30678. Other data that support the findings of this study are available from the corresponding authors upon reasonable request.

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

## Acknowledgements

We thank Stephen Harrison for insightful discussion and critical reading of the manuscript. Alexander Daniels and Catharina Saenger kindly helped with protein construct design and testing. Zhijun Liu helped with NMR data collection at NCPSS. This work was supported by NIH grants AI127193 (to B.C. and J.J.C.), GM116898 (to J.J.C.), AI129721 (to B.C.), a Merck Fellowship (to Q.F.), and Collaboration for AIDS Vaccine Discovery (CAVD) grant OPP1169339 (to Dan H. Barouch from the Bill and Melinda Gates Foundation). The NMR data were collected at the MIT-Harvard CMR (supported by NIH grant P41 EB-002026).

## Author contributions

A.P., B.C., and J.J.C. conceived the study; A.P. designed protein constructs for structural studies; Q.F. designed protein constructs for NMR dynamics studies; A.P. and Q.F. prepared samples for NMR studies; A.P., Y.C., B.C., and J.J.C. designed mutants for functional studies; F.G., Y.C., T.X., M.M.S., H.P., S.R.-V., and M.S.S. performed functional studies; W.C. and A.P. performed OG-label analysis; A.P. and J.J.C. solved the NMR structure; A.P., B.C., and J.J.C. wrote the paper and all authors contributed to editing of the manuscript.

## Competing interests

The authors declare no competing interests.
