## [Peer Review File · Nature Communications]

Reviewers' comments:

Reviewer #1 (Remarks to the Author):

In their manuscript entitled "Structural Basis of Transmembrane Coupling of the HIV-1 Envelope Glycoprotein" Xiao et al present the NMR solution structure of the transmembrane region (TMD and CT) of the HIV Env protein. They show that the C-terminal membrane-proximal region is forming a so-called "baseplate" that is supporting the TMDs and which is possibly modulating the dynamics of the MPER region and ultimately the ectodomains that bind to host cells and mediate HIV infectivity.

This manuscript covers a very relevant biological topic and provides important structural information on the transmembrane region of Env that is not easily accessible via X-ray crystallography or cryoEM. The herein employed structure determination procedure follows in principle a protocol that has been established by the Chou lab published this year. The authors use differential isotope labeling to selectively extract inter-subunit NOEs, as well as a variety of PRE restraints that help define the membrane position of the C-terminal LLP2 region. The PRE effects have been calibrated by using the previously in the same lab determined structure of the TMD. It would be informative and convincing to include prototype inter-subunit NOE data in Fig. 1 in order to visualize the data quality of essential NOE restraints that have been used for structure calculation.

The authors mention that bicelles represent a real bilayer environment. However, a fair amount of detergent is still present in such samples with $q=0.5$. Have the authors also explored other membrane mimicking environments, such as lipid nanodiscs, and/or other lipids that might be a better mimic of a biological membrane?

The authors verified the obtained structure with binding assays with TMD and CT variants and a large set of antibodies that bind to the trimeric form of Env or the CD4 binding region. However, a more direct detection of the effect of a few critical mutations by NMR would add further confidence to the presented structure, e.g. monitoring the effect of spin-labeled and mutated CT on wt TMD. For these PRE experiments a control with spin label only would be appropriate to assay the effect of unspecific incorporation of the spin label into bicelles, which would also affect the signal intensities of the TMD.

I think the epitopes of the antibodies need to be described in more detail in order to understand the specific effects shown in Fig. 3. Also, the authors report on altered binding of a trimer-specific antibody with these TMD and CT variants. However, since the TMD is a trimer already, how would a (partial) disruption of the TMD-CT interaction lead to decreased trimerization of the TMD? And, can the increased infectivity of the F774N variant be explained with the structure and the presented baseplate model?

NMR dynamics: The locking of the MPER region dynamics by inter-chain disulfide bond formation is very elegant, even though a reduction of internal dynamics is somewhat expected for such a species. The TMD-CT construct is a bit less mobile than the MPER linked construct at the C-terminal half of the TMD, even though a CT baseplate should restrict dynamics by direct binding to the TMD. This issue requires some explanation. Furthermore, since the dynamics and conformation of the MPER region seems to be essential for the mechanism of ectodomain repositioning presented in Fig. 4b, it would be interesting and mechanistically insightful to compare the dynamics of MPER in the MPER-TMD and MPER-TMD-CT constructs. The dynamics of MPER should be markedly reduced if the CT is present, providing more direct prove for the proposed model. In addition, spin labels at the MPER region would provide information on possible structural changes within MPER in MPER-TMD versus MPER-TMD-CT constructs.

In Fig. 4b, the authors mention that CT baseplate needs to dissociate or loosen its interaction with

the TMD to induce ectodomain perturbations. Even though a direct verification is difficult, this point needs to be addressed in the discussion. Are there partner proteins that might induce that process or posttranslational modifications, ...?

In summary, this paper presents a beautiful structure of the HIV Env TMD and the C-terminal LLP2 region that so far could not be resolved by other structural methods. The structure determination procedure is accurate and based on various experimental restraints. However, the functional relevance of the TMD-baseplate interaction on MPER dynamics and conformation needs to be shown in more detail by additional NMR experiments.

Reviewer #2 (Remarks to the Author):

Piai et al. present a structure of part of the cytoplasmic tail (CT) and the transmembrane domain (TMD) of the HIV-1 envelope glycoprotein trimer. Based on the structure the authors introduce a number of mutations in the CT that influence the antigenic structure of the Env ectodomain. This supports the longstanding notion that the CT has an impact on the structure of the ectodomain.

I have no problems with the finding that the cytoplasmic tail influences the ectodomain. This is an important finding and should be published, although the supporting data can and should be strengthened. The interpretation in terms of ectodomain structures and vaccine design are more problematic and should be revised.

Major concerns:

1. While the major conclusion that cytoplasmic tail has effects on the Env ectodomain are valid based on the phenotypic assays, the characterization and interpretation could be strengthened. Most single mutations have subtle effects on antigenicity, while several combinations of mutations have strong phenotypic effects. It appears that these latter viruses appear to have a tier-1 virus-resistance phenotype as exemplified by high sensitivity to b6 and 3791 and resistance to V2-apex bnAbs, usually associated with an open ectodomain structure. This should be probed further. The most important pseudoviruses should be tier-categorized using polyclonal sera panels commonly used for this purpose and the overall sensitivity should be plotted. If my assumption that these combinations of mutations converted a tier-2 virus into a tier-1 virus is correct, then the interpretation of the data and the relevance to vaccine design (see also next point) should be revisited because the data in this paper then lend strong support that the currently known high-resolution structures, all competent in binding V2-apex bnAbs (some even solved in complex with such bnAbs) do represent appropriate mimics of the native Env trimer. In addition to tier-categorization, the authors might want to strengthen the phenotypic data by inserting selected (combinations) of mutations to other virus strains, in particular those from clades B and C. The phenotyping and structural work were performed using different virus strains. The authors observe differences with three V2-apex bnAbs (and reverse effects with some non-NABs), but not with the VRC01 control bnAb. The data should be substantiated with multiple bnAbs from different clusters, in any case with some selected combination mutants. Which epitope clusters are affected by modification of CT and which ones are not? Presumably only the V2-apex is affected, but that is just an assumption. The quaternary structure dependent epitopes at the gp120/gp41 interface could also be affected.

2. The authors overemphasize the relevance of the new structures for vaccine design (Lines 263-265). It is unclear how this new information informs vaccine design directly. If anything, this study supports the use of the native-like trimer platforms currently used as these are compatible with V2-apex bnAb binding. Furthermore, the existing structures are not controversial (Line 259-263) and are all in agreement with one another, whether they are of SOSIP trimers, NFL trimers, or unmodified native membrane-derived Env. On line 60 and also in the discussion the authors need

to refer to (and discuss) two recent structures of native, membrane-derived Env trimers (doi: 10.1371/journal.ppat.1007920; doi.org/10.1101/730333). These structures are relevant in several ways. First, they lend further support to the notion that the existing high-resolution structures are representative for the native Env trimer. Second, one of them uses the same isolate as is used here for the phenotypic assays. Third, one of them was obtained with a construct that included the CT (although the CT was not resolved). Fourth and importantly, they include two structures of native membrane-derived Env in complex with V2-apex directed bnAbs, including ones that the authors have studied here and for which effects were observed in the phenotypic assays as a result of CT modification. In fact, one of the senior authors of this paper is also an author on one of these papers (doi.org/10.1101/730333) so its omission is somewhat odd. The conclusions are also somewhat contradictory to the conclusions drawn here. In that paper the authors (including one of the senior authors of this paper) concluded that the existing structures provide good pictures of the Env ectodomain on virions and that the interpretation of smFRET data are inaccurate. This reviewer agrees and the data presented in this manuscript further support the relevance of the existing ectodomain structures for vaccine design as they are able to interact with bNAbs against the V2-apex. Given the above, it is appropriate that the authors present a structural model of the CT, TMD as well as the complete ectodomain and discuss this further.

Minor concerns:

Line 40. "than previously appreciated" This is not true. The role of the CT in the antigenic conformation of the ectodomain has been known for more than 25 years. See for example ref 15 and the references therein.

Line 85. What is the expected effect of removing the palmitoylation anchor? Furthermore, is residue 764 associated or close to the membrane? The location of this residue might serve as an indicator whether the structure is correct as this residue is expected to be in close contact with the membrane.

Line 96. "CT was properly folded". This cannot be deduced from these data, only that CT did not disrupt TMD.

Line 157. The rationale for generating the individual mutations as well as the particular combinations of mutations should be given.

Line 259. "other strain-specific" does not make sense. Mimics of the prefusion trimer are expected (and do) induce strain-specific nAbs. I.e. a native-like trimer of a given strain is expected to induce nAbs against that particular strain. The induction of strain-specific nAbs is not a consequence of nonnative structure.

Response to Reviewers

Reviewer #1:

In their manuscript entitled “Structural Basis of Transmembrane Coupling of the HIV-1 Envelope Glycoprotein” Piai et al present the NMR solution structure of the transmembrane region (TMD and CT) of the HIV Env protein. They show that the C-terminal membrane-proximal region is forming a so-called “baseplate” that is supporting the TMDs and which is possibly modulating the dynamics of the MPER region and ultimately the ectodomains that bind to host cells and mediate HIV infectivity.

This manuscript covers a very relevant biological topic and provides important structural information on the transmembrane region of Env that is not easily accessible via X-ray crystallography or cryoEM. The herein employed structure determination procedure follows in principle a protocol that has been established by the Chou lab published this year. The authors use differential isotope labeling to selectively extract inter-subunit NOEs, as well as a variety of PRE restraints that help define the membrane position of the C-terminal LLP2 region. The PRE effects have been calibrated by using the previously in the same lab determined structure of the TMD.

It would be informative and convincing to include prototype inter-subunit NOE data in Fig. 1 in order to visualize the data quality of essential NOE restraints that have been used for structure calculation.

The reviewer may have missed the Supplementary Figure 4 which shows the prototypical inter-subunit NOE as well as schematic illustration of the J(CH)-modulated NOE experiment for obtaining the inter-subunit NOEs. Since this is a rather big technical figure, we believe it is more suitable as supplementary data.

The authors mention that bicelles represent a real bilayer environment. However, a fair amount of detergent is still present in such samples with $q=0.5$. Have the authors also explored other membrane mimicking environments, such as lipid nanodiscs, and/or other lipids that might be a better mimic of a biological membrane?

Good point. Indeed, for DMPC-DHPC bicelles with $q = 0.5$, there is about 5 mM free DHPC, which could have some effect on the protein, especially on the soluble, extramembrane domains. For this consideration, we have attempted to study the membrane-related components of HIV-1 Env in lipid nanodiscs, but only managed to generate a reasonable NMR spectrum for the transmembrane domain (TMD) in the nanodisc (unpublished). As shown below (Fig. 1), the HSQC spectra of the TMD in bicelles ($q = 0.5$) and nanodisc are very similar, suggesting that the structures in the two environments are similar. We were not able to generate feasible NMR sample of the MPER-TMD or TMD-CT in nanodisc, possibly due to interaction of the amphipathic helices with those of the nanodisc scaffold protein MSP.

Figure 1. NMR spectra of the Env TMD in bicelle and in lipid nanodisc. The ^1H - ^{15}N TROSY-HSQC spectra of gp41^{HIV1D}(677-716) trimer reconstituted in bicelle with (^{15}N , ^1H)-labeled protein (left), bicelle with (^{15}N , ^2H)-labeled protein (middle), and DMPC nanodisc with (^{15}N , ^1H)-labeled protein (right).

Despite the presence of detergent, the DMPC-DHPC bicelles with $q \geq 0.5$ should still very closely mimic a lipid bilayer. NMR and SAXS studies of bicelles at various q values have independently shown that $q = 0.5$ is the critical point at which the DMPC-DHPC bicelles transition from mixed micelles to bicelles with lipid disc shape (Piai et al, *Chemistry*, 23:1361-1367, 2017; Caldwell et al, *J Phys Chem Lett*, 9:4469-4473, 2018). Since the TMD-CT construct is mostly buried in bicelles, as our paramagnetic probe titration data indicated, the small amount of free DHPC should not have a significant effect on the TMD-CT structure.

The authors verified the obtained structure with binding assays with TMD and CT variants and a large set of antibodies that bind to the trimeric form of Env or the CD4 binding region. However, a more direct detection of the effect of a few critical mutations by NMR would add further confidence to the presented structure, e.g. monitoring the effect of spin-labeled and mutated CT on wt TMD. For these PRE experiments a control with spin label only would be appropriate to assay the effect of unspecific incorporation of the spin label into bicelles, which would also affect the signal intensities of the TMD.

We thank the reviewer for suggesting such an interesting experiment. To evaluate the structural impact of weakening the interaction between the CT and TMD, we introduced five mutations in the CT H1 including L748S, L755S, D758A, D759A, and S762A. The conformational stability of the mutant (designated CT2-tmd) was examined by inter-chain PRE analysis. As previously done for the TMD-CT^{LLP2} (Supplementary Fig. 6b), we prepared a mixed sample containing ~1:1 (^{15}N , ^2H)-labeled CT2-tmd and unlabeled CT2-tmd carrying the spin-label at S764C (in the H1 helix of the CT), and measured residue-specific PRE (I/I_0). Comparison of residue-specific PREs to that measured under identical condition for the wildtype TMD-CT^{LLP2} shows that the TMD PRE values of the mutant are reduced by as much as 30% (see the new Fig. 3 in the revised manuscript). Since the two samples carry the same spin-label at C764, the results

indicate that the H1 mutations above indeed have weakened the CT-TMD interaction and probably loosened the CT baseplate.

We also agree with the reviewer that complete removal of free MTSL is critical for the PRE analysis. In fact, we had already included such a negative control in the original manuscript (see METHODS and Supplementary Fig. 6d). In that experiment, a sample containing only (¹⁵N, 85% ²H)-labeled TMD-CT^{LLP2} (without Cys) was prepared using the same MTSL-labeling procedure used for the mixed samples. The result shows that the PRE vs. Residue Number plot is essentially flat at ~1, indicating that free MTSL was completely removed.

I think the epitopes of the antibodies need to be described in more detail in order to understand the specific effects shown in Fig. 3. Also, the authors report on altered binding of a trimer-specific antibody with these TMD and CT variants. However, since the TMD is a trimer already, how would a (partial) disruption of the TMD-CT interaction lead to decreased trimerization of the TMD? And, can the increased infectivity of the F774N variant be explained with the structure and the presented baseplate model?

This is an excellent question which we intended to address with the mechanistic drawing in Fig. 4b (Fig. 5b in the revised manuscript). First, it is important to clarify here that “destabilization of the TMD trimer”, a phrase we used in our previous papers on the TMD, does not necessarily mean dissociation of the TMD trimer. In fact, in an earlier H-D exchange study (Piai et al, *JACS*, 139(51):18432-18435, 2017), we found that the TM helices are held together extremely strongly at the hydrophobic core (residues 686-689) in the N-terminal half of the TMD, whereas the C-terminal half does not show strong helix-helix packing and is subject to substantial ms-us dynamics. These results prompted the notion that the TMD trimer, if separated from the rest of the Env, could undergo a “scissor-like” movement around the hydrophobic core (or the hinge), and that constraining the C-terminal end of the TMD with CT could allosterically constrain the movements of the MPER at the N-terminal end, and vice versa. Hence, in our model (Fig. 5b in the revised manuscript), disruption of the TMD-CT interaction does not cause decreased TMD trimerization; it loosens the structural constraints at the C-terminal end of the TMD such that the TMD undergoes greater “scissor” motion, causing the MPER, and consequently the Env, to be more open. This model was at least partially vindicated by the NMR dynamics experiment showing that locking the MPER at the N-terminal end of the TMD significantly reduced the movement at its C-terminal end (see response to the related question below).

NMR dynamics: The locking of the MPER region dynamics by inter-chain disulfide bond formation is very elegant, even though a reduction of internal dynamics is somewhat expected for such a species. The TMD-CT construct is a bit less mobile than the MPER linked construct at the C-terminal half of the TMD, even though a CT baseplate should restrict dynamics by direct binding to the TMD. This issue requires some explanation.

We believe the reviewer probably meant the TMD-CT construct is a bit *more* mobile than the MPER linked construct at the C-terminal half of the TMD. It is true that locking

the MPER is expected to result in reduced protein internal dynamics, but only for the MPER region itself where the inter-chain disulfides are introduced; it is still quite remarkable that this effect is propagated to the C-terminal end of the TMD ~50 residues away. This observation suggests that the proposed cross-membrane coupling of the Env is entirely possible by structural consideration. Regarding the issue of the TMD C-terminal end showing more dynamics in the TMD-CT than the locked MPER-TMD, we believe this result is not inconsistent with the proposed model. Comparing to the unlocked MPER-TMD, the TMD-CT construct did show reduced dynamics at the C-terminal end of the TMD. But the CT-TMD interaction still may not be as strong as covalently locking the MPER. The key message from the NMR dynamics study is that the pivotal or the scissor-like motion of the TMD can be modulated by either MPER or CT.

Furthermore, since the dynamics and conformation of the MPER region seems to be essential for the mechanism of ectodomain repositioning presented in Fig. 4b, it would be interesting and mechanistically insightful to compare the dynamics of MPER in the MPER-TMD and MPER-TMD-CT constructs. The dynamics of MPER should be markedly reduced if the CT is present, providing more direct prove for the proposed model. In addition, spin labels at the MPER region would provide information on possible structural changes within MPER in MPER-TMD versus MPER-TMD-CT constructs.

We agree with the Reviewer that the MPER-TMD-CT^{LLP2} construct is extremely useful as it would allow us to directly examine whether the presence of the CT can reduce the dynamics of the MPER. Unfortunately, we have attempted to express this construct for a long time but to no avail! We eventually reconciled with the locked MPER-TMD to look at coupling in the MPER to CT direction.

In Fig. 4b, the authors mention that CT baseplate needs to dissociate or loosen its interaction with the TMD to induce ectodomain perturbations. Even though a direct verification is difficult, this point needs to be addressed in the discussion. Are there partner proteins that might induce that process or posttranslational modifications, ...?

When stating that loosening or dissociation of the CT baseplate can induce perturbations to the ectodomain, we did not mean this process or phenomenon is relevant to viral entry. The structural results in this paper provide a structural basis for explaining the intriguing coupling between the Env ectodomain and CT observed previously (Chen et al, Science, 349:191-195, 2015), and this has important implication to immunogen design for HIV vaccine development. We have made this point clear in the discussion.

Reviewer #2:

Piai et al. present a structure of part of the cytoplasmic tail (CT) and the transmembrane domain (TMD) of the HIV-1 envelope glycoprotein trimer. Based on the structure the

authors introduce a number of mutations in the CT that influence the antigenic structure of the Env ectodomain. This supports the longstanding notion that the CT has an impact on the structure of the ectodomain.

I have no problems with the finding that the cytoplasmic tail influences the ectodomain. This is an important finding and should be published, although the supporting data can and should be strengthened. The interpretation in terms of ectodomain structures and vaccine design are more problematic and should be revised.

Major concerns:

1. While the major conclusion that cytoplasmic tail has effects on the Env ectodomain are valid based on the phenotypic assays, the characterization and interpretation could be strengthened. Most single mutations have subtle effects on antigenicity, while several combinations of mutations have strong phenotypic effects. It appears that these latter viruses appear to have a tier-1 virus-resistance phenotype as exemplified by high sensitivity to b6 and 3791 and resistance to V2-apex bnAbs, usually associated with an open ectodomain structure. This should be probed further. The most important pseudoviruses should be tier-categorized using polyclonal sera panels commonly used for this purpose and the overall sensitivity should be plotted. If my assumption that these combinations of mutations converted a tier-2 virus into a tier-1 virus is correct, then the interpretation of the data and the relevance to vaccine design (see also next point) should be revisited because the data in this paper then lend strong support that the currently known high-resolution structures, all competent in binding V2-apex bnAbs (some even solved in complex with such bnAbs) do represent appropriate mimics of the native Env trimer. In addition to tier-categorization, the authors might want to strengthen the phenotypic data by inserting selected (combinations) of mutations to other virus strains, in particular those from clades B and C. The phenotyping and structural work were performed using different virus strains. The authors observe differences with three V2-apex bnAbs (and reverse effects with some non-NAbs), but not with the VRC01 control bnAb. The data should be substantiated with multiple bnAbs from different clusters, in any case with some selected combination mutants. Which epitope clusters are affected by modification of CT and which ones are not? Presumably only the V2-apex is affected, but that is just an assumption. The quaternary structure dependent epitopes at the gp120/gp41 interface could also be affected.

We thank the reviewer for the excellent suggestion. We have now performed tier phenotyping using 7 HIV+ chronic serum samples for the 92UG037.8 (a tier 2 virus) WT and the TMD-ct mutant, along with the CT2-tmd and CT3-tmd mutants as representative “intermediate” phenotype viruses (Supplementary Table 3 in the revised manuscript). As predicted by the reviewer, the TMD-ct mutant showed a higher level of sensitivity to almost all 7 HIV+ chronic serum samples than the WT, consistent with an open Env conformation and the tier-1 phenotype. The two intermediate mutants are similar to the WT in tier phenotyping, in agreement with our interpretation that these mutations only lead to limited local changes next to the trimer apex.

We have also tested additional bnAbs that target CD4bs, V3-glycan, V1/V2-glycan, gp120/gp41 interface, and the MPER, as suggested (Supplementary Table 4 in the revised manuscript). The WT and mutant viruses showed the same sensitivities to most of the bnAbs against the CD4bs and V3-glycan epitopes. All the mutants are slightly less sensitive to the V1/V2-glycan bnAb PGDM1400 and the gp120/gp41 interface bnAb 8ANC194, but they demonstrated ~10-fold higher sensitivity to the MPER antibody 4E10, suggesting that the mutations in the CT and/or TM regions at least destabilize the MPER structure.

We totally agree with the reviewer that it is important to introduce the same mutations into other virus strains, in particular those from clades B and C, to recapitulate the phenotypes observed in the 92UG037.8 (clade A) virus. We have indeed ordered synthetic genes for two other Envs (C97ZA012, clade C and CH120.6, a chronic circulating recombinant form (CRF) 07_BC) with the mutations, but we were recently informed by Genscript Biotech that there would be major delays for all other projects except for those related to the 2109-nCoV outbreak. For the moment, we are unable to predict when the experiment can be finished since we will have to test the expression of the mutant Env constructs once available, and generate and characterize pseudoviruses before testing antibody sensitivity. At the same time, we feel that we have already provided some level of cross-clade validation because the structural predictions were made based on an NMR structure using a sequence derived from a clade D isolate 92UG024.2, and yet the phenotype analysis was all performed with a clade A isolate 92UG037.8. We therefore believe these data are not essential for our main conclusion in our manuscript.

2. The authors overemphasize the relevance of the new structures for vaccine design (Lines 263-265). It is unclear how this new information informs vaccine design directly. If anything, this study supports the use of the native-like trimer platforms currently used as these are compatible with V2-apex bNAb binding. Furthermore, the existing structures are not controversial (Line 259-263) and are all in agreement with one another, whether they are of SOSIP trimers, NFL trimers, or unmodified native membrane-derived Env. On line 60 and also in the discussion the authors need to refer to (and discuss) two recent structures of native, membrane-derived Env trimers (doi: 10.1371/journal.ppat.1007920; doi.org/10.1101/730333). These structures are relevant in several ways. First, they lend further support to the notion that the existing high-resolution structures are representative for the native Env trimer. Second, one of them uses the same isolate as is used here for the phenotypic assays. Third, one of them was obtained with a construct that included the CT (although the CT was not resolved). Fourth and importantly, they include two structures of native membrane-derived Env in complex with V2-apex directed bnAbs, including ones that the authors have studied here and for which effects were observed in the phenotypic assays as a result of CT modification. In fact, one of the senior authors of this paper is also an author on one of these papers (doi.org/10.1101/730333) so its omission is somewhat odd. The conclusions are also somewhat contradictory to the conclusions drawn here. In that paper the authors (including one of the senior authors of this paper) concluded that the existing structures provide good pictures of the Env ectodomain on virions and that the

interpretation of smFRET data are inaccurate. This reviewer agrees and the data presented in this manuscript further support the relevance of the existing ectodomain structures for vaccine design as they are able to interact with bNAbs against the V2-apex. Given the above, it is appropriate that the authors present a structural model of the CT, TMD as well as the complete ectodomain and discuss this further.

We totally understand the reviewer's point of view and are happy to tone down the discussion on the relevance of the new structures for vaccine design since the HIV vaccine community as a whole does not have a good understanding of structural correlates with effective immunogenicity. Our data suggest strategies to stabilize or modulate the antigenic structure of the Env ectodomain without introducing any mutations in it, but we cannot guarantee these approaches would lead to an immunogen that induces bnAbs. After all, the native-like trimers, such as SOSIP, have yet to induce any durable and potent bnAbs responses which are likely required for a successful vaccine, even just in animal models.

We have cited the two structural studies of the membrane-derived Env trimers in the revised manuscript, as suggested (yes, the reviewer is correct that Dr. Bing Chen is a co-author of one of the published studies). However, we also want to emphasize that the two studies used the full-length Env trimers but both solubilized in detergent micelles. It is well documented that detergent micelles cannot mimic a lipid bilayer (a real membrane) in many aspects. For example, the MPER structures studied in detergent are monomeric, while the MPER together with the TMD reconstituted in lipid bilayer is trimeric and more physiologically relevant (Fu et al. PNAS, 2018). Therefore, the question remains – do the disordered regions (the MPER, TMD and CT) in those two intact Env structures really represent their conformation in the native Env trimer on the surface of virion? The answer is probably NOT based on many biological observations reported in literature. This kind of uncertainty was the main reason we did not cite those studies because we did not know, for the moment, how to reconcile the current study and those two studies. Nevertheless, we are actively pursuing the cryoEM structure of the full-length Env trimer reconstituted in lipid-based nanodiscs and we may have a far better interpretation once that structure becomes available.

What we meant by “controversial” is the discrepancy between the existing structures and the smFRET data. As much as we would like to agree with the reviewer that the existing high-resolution structures are representative for the native Env trimer, but there is no convincing evidence or explanation that can totally dismiss the observations of Env on the surface of virion by smFRET, and each of the existing structure has its own limitations. We believe that more data are needed to draw a conclusion about Env structure relevant to immunogen design one way or the other.

We have indeed included a model of the entire Env including the ectodomain and the membrane-related components is presented in Fig. 1e. The model was obtained by fitting the MPER-TMD-CT^{LLP2} structure and the structure of the SOSIP Env trimer into the low-resolution EM density of the HIV-1 Env trimer on the virion surface by cryo-electron tomography. Although the critical structural information of the connection

between the ectodomain and the MPER is missing, the model, as it appears, would be consistent with conformational coupling between the ectodomain and the CT of the Env via interaction between the CT and the TMD (schematically illustrated in Fig. 5b of the revised manuscript).

Minor concerns:

Line 40. "than previously appreciated" This is not true. The role of the CT in the antigenic conformation of the ectodomain has been known for more than 25 years. See for example ref 15 and the references therein.

The reviewer is right that there were early reports suggesting that mutations in the CT could affect the antigenic properties of Env, but most of those studies relied on Envs that are not homogeneous, in particular, those from lab-adapted, tier 1 isolates. With more understanding of properties of Envs from different isolates, we need to be cautious when trying to interpret those data. Nevertheless, we have removed the phrase in the revised manuscript.

Line 85. What is the expected effect of removing the palmitoylation anchor? Furthermore, is residue 764 associated or close to the membrane? The location of this residue might serve as an indicator whether the structure is correct as this residue is expected to be in close contact with the membrane.

Excellent point! The Ser764 sidechain indeed points to the bilayer core (see Fig. 1c in the revised manuscript) and modeling of palmitoylation at this residue indicates proper partitioning of the acyl chain in the bilayer, as shown below (Fig. 2). This has been emphasized in the revised manuscript.

Figure 2. Modeling of palmitoylation at C764 (S764 in the TMD-CT^{LLP2} construct) showing that the palmitoylation sites all point towards the lipid bilayer core.

Line 96. "CT was properly folded". This cannot be deduced from these data, only that CT did not disrupt TMD.

We agree with the reviewer and have removed this claim in the revised version.

Line 157. The rationale for generating the individual mutations as well as the particular combinations of mutations should be given.

We thank the reviewer for the suggestion and we have added explanation for why the mutations were generated.

Line 259. "other strain-specific" does not make sense. Mimics of the prefusion trimer are expected (and do) induce strain-specific nAbs. I.e. a native-like trimer of a given strain is expected to induce nAbs against that particular strain. The induction of strain-specific nAbs is not a consequence of nonnative structure.

The reviewer is right that there are many terms used in the HIV research community are neither sensible nor accurate. For example, "native" trimer often means an Env trimer with antigenic properties mimicking the viral Env on the surface of virion and an Env sampling the open conformation with non-neutralizing epitopes exposed is often considered as "non-native", but it is fully functional, like those derived from tier-1 viruses. From the protein structure point of view, one should not call a functional protein non-native. Due to the historical reason, the term "strain-specific" is used to refer those antibodies that can potently neutralize certain strains (not just one particular strain), especially, the tier-1 isolates, but not other tier-2 or tier-3 isolates, which are more relevant to vaccine development, in order to distinguish with those broadly neutralizing antibodies. Some investigators choose to use "narrowly neutralizing" instead. We are happy to rephrase if both the reviewer and editor feel strongly about it.

REVIEWERS' COMMENTS:

Reviewer #1 (Remarks to the Author):

In their revised manuscript entitled "Structural Basis of Transmembrane Coupling of the HIV-1 Envelope Glycoprotein" Xiao et al have provided additional data that verified the structural model presented in the manuscript. The authors have clarified most of my initial concerns by either the addition of new data to the manuscript or unpublished data shown in the rebuttal letter.

I just have a few points to be considered for publication:

The antibody assay data are difficult to understand for the general reader. For better clarity, the authors might want to add a bit more information why these 7 different antibodies have been used and what epitopes are recognized.

It is surprising that locking the MPER region by disulfide bridges has a stronger effect on the dynamics of the C-terminal end of the TMD than direct interaction with the CT. The authors explain this issue in their letter. However, this should be added to the results or discussion section, too.

In addition, looking at the relaxation data, I wonder whether the TMD structure presented in an earlier publication by the authors (Ref 8) is less well defined at its C-terminal end (e.g. line broadening effects and missing NOE contacts) than the TMD-CT structure presented here?

It is unfortunate that a MPER-TMD-CT construct is not available, which should, according to the presented model, show reduced dynamics at the MPER region. I wonder whether such data can be generated with an MPER-TMD construct instead (which can be produced) and by addition of the (non-isotope-labeled) CT produced separately, similar to what the authors used for their PRE experiments? This would render the dynamics part of the paper more convincing.

Reviewer #2 (Remarks to the Author):

I am happy with the addition of the tier categorization of the mutants and the analyses with additional antibodies. Both sets of data strengthen the functional aspects of the paper.

I am also happy that the authors have made some modifications to the discussions. I do disagree with the authors' statement that "there is no convincing evidence or explanation that can dismiss the observations by smFRET". The authors should take a look at a paper from the Bjorkman lab studying Env dynamics by a different and arguably more sophisticated technique than smFRET: DEER spectroscopy (PMID:30076100). While the DEER paper only addresses soluble Env trimers, not virion associated trimers, the discrepancies between DEER and smFRET performed on soluble trimers, seriously compromises the distinction between states 1 and 2 as concluded from smFRET. One can also question why cryo-EM analyses on SOSIP trimers, which according to the smFRET data occupies both states 1 and 2, have never shown two distinct states. The most likely explanation is that the interpretation of the smFRET data is based on artefacts. As too why smFRET might lead to artefactual conclusions on Env structure, I again refer the authors to the paper by Stephen Harrison, mentioned in my previous review (PMID:31931014). Thus, work on Env structure, biophysics, dynamics (DEER), and antigenicity, all converge to the same conclusions, while the smFRET data are highly controversial.

Finally, I agree with the authors that there is some ambiguity in the field with respect to the terminology of strain-specific nAbs, non-nAbs, narrow-specificity nAbs etc. and I do now see why the authors used the term here. If the authors are indeed referring to tier 1 nAbs (which are usually non-nAbs for tier 2 viruses and usually dominated by V3-specificities), I would recommend

using the term "tier 1 nAbs", as "strain-specific nAbs" could be confused with autologous tier 2 nAbs for example.

Response to Reviewers

Reviewer #1:

In their revised manuscript entitled “Structural Basis of Transmembrane Coupling of the HIV-1 Envelope Glycoprotein” Xiao et al have provided additional data that verified the structural model presented in the manuscript. The authors have clarified most of my initial concerns by either the addition of new data to the manuscript or unpublished data shown in the rebuttal letter.

I just have a few points to be considered for publication:

1. The antibody assay data are difficult to understand for the general reader. For better clarity, the authors might want to add a bit more information why these 7 different antibodies have been used and what epitopes are recognized.

Good suggestion. The 7 antibodies used are the well-established neutralizing/non-neutralizing markers for evaluating the openness of the HIV-1 Env ectodomain. We have included a Supplementary Fig. 10 to show the mapping of the most important epitopes.

2. It is surprising that locking the MPER region by disulfide bridges has a stronger effect on the dynamics of the C-terminal end of the TMD than direct interaction with the CT. The authors explain this issue in their letter. However, this should be added to the results or discussion section, too.

Good suggestion. We have included a paragraph in the Discussion for explaining the above intriguing observation.

3. In addition, looking at the relaxation data, I wonder whether the TMD structure presented in an earlier publication by the authors (Ref 8) is less well defined at its C-terminal end (e.g. line broadening effects and missing NOE contacts) than the TMD-CT structure presented here?

A qualitative comparison of residue-specific R_1 and R_2 profiles between TMD (measured previously under slightly different conditions, i.e., 303 K) and TMD-CT^{LLP2} does not show obvious differences. But, TMD alone is probably not a good construct for testing the MPER – CT coupling hypothesis because the presence of MPER, which exhibits substantial dynamics, should strongly affect the dynamics of the TMD. Indeed, this is what we saw in the difference between the locked and unlocked MPER-TMD (Fig. 5a). Therefore, the ultimate way to directly look at the MPER – CT coupling by NMR is to have the MPER-TMD-CT^{LLP2} construct. Unfortunately, we are still struggling with this construct right now.

It is unfortunate that a MPER-TMD-CT construct is not available, which should, according to the presented model, show reduced dynamics at the MPER region. I

wonder whether such data can be generated with an MPER-TMD construct instead (which can be produced) and by addition of the (non-isotope-labeled) CT produced separately, similar to what the authors used for their PRE experiments? This would render the dynamics part of the paper more convincing.

Thank you for bringing this up! We had thought about the suggested experiment but decided that such experiment would take just as much optimization effort and time as achieving the MPER-TMD-CT construct while unlikely to generate convincing results. First, our MPER-TMD fragment (ending at TM residue Q710) was derived previously from extensive screen of constructs of various length. For unknown reasons, only the MPER-TMD fragment 660-710 expressed well enough for NMR samples (Fu et al, PNAS 2018). But, R709 and Q710, both important for interacting with the CT (see Fig. 1b), are disordered due to end truncation. Hence, the available MPER-TMD construct will probably not interact strongly with the CT^{LLP2}. In the referenced PRE experiment in Fig. 2, we mixed the CT^{LLP2} with the TMD-KS fragment that contains a very significant stretch of native sequence after the TMD. Second, when separated, the TMD interacts with the CT^{LLP2} much more weakly than in the full-length construct (see OG-label analyses in Supplementary Figs. 9b and 1b), thus cannot adequately recapitulate the MPER – CT coupling.

Reviewer #2:

I am happy with the addition of the tier categorization of the mutants and the analyses with additional antibodies. Both sets of data strengthen the functional aspects of the paper.

We thank the reviewer for the very supportive comments.

I am also happy that the authors have made some modifications to the discussions. I do disagree with the authors' statement that "there is no convincing evidence or explanation that can dismiss the observations by smFRET". The authors should take a look at a paper from the Bjorkman lab studying Env dynamics by a different and arguably more sophisticated technique than smFRET: DEER spectroscopy (PMID:30076100). While the DEER paper only addresses soluble Env trimers, not virion associated trimers, the discrepancies between DEER and smFRET performed on soluble trimers, seriously compromises the distinction between states 1 and 2 as concluded from smFRET. One can also question why cryo-EM analyses on SOSIP trimers, which according to the smFRET data occupies both states 1 and 2, have never shown two distinct states. The most likely explanation is that the interpretation of the smFRET data is based on artefacts. As too why smFRET might lead to artefactual conclusions on Env structure, I again refer the authors to the paper by Stephen Harrison, mentioned in my previous review (PMID:31931014). Thus, work on Env structure, biophysics, dynamics (DEER), and antigenicity, all converge to the same conclusions, while the smFRET data are highly controversial.

We notice that the reviewer disagreed with a statement in our previous rebuttal letter. We have indeed carefully studied the DEER paper by Dr. Pamela Bjorkman even before this manuscript and have also had extensive discussion on the subject with her. Likewise, we thanked Dr. Stephen Harrison for insightful discussion and critical reading of the current manuscript. We appreciate the reviewer's view about the highly complex HIV literature and have cited the paper by Dr. Bjorkman in the revised manuscript, as suggested.

Finally, I agree with the authors that there is some ambiguity in the field with respect to the terminology of strain-specific nAbs, non-nAbs, narrow-specificity nAbs etc. and I do now see why the authors used the term here. If the authors are indeed referring to tier 1 nAbs (which are usually non-nAbs for tier 2 viruses and usually dominated by V3-specificities), I would recommend using the term "tier 1 nAbs", as "strain-specific nAbs" could be confused with autologous tier 2 nAbs for example.

We thank the reviewer for the excellent suggestion. We have replaced "strain-specific nAbs" with "tier 1 nAbs", as suggested.